# Lactate supports cell-autonomous ECM production to sustain metastatic behavior in prostate cancer

Luigi Ippolito [ID] [1,4 ✉], Assia Duatti [1,4], Marta Iozzo [ID] [1], Giuseppina Comito [1], Elisa Pardella [ID] [1], Nicla Lorito [1], Marina Bacci [ID] [1], Erica Pranzini [ID] [1], Alice Santi [1], Giada Sandrini [ID] [2], Carlo V Catapano [ID] [2], Sergio Serni [3], Pietro Spatafora [ID] [3], Andrea Morandi [ID] [1], Elisa Giannoni [ID] [1,4] & Paola Chiarugi [ID] [1,4 ✉]

## Abstract

Extracellular matrix (ECM) is a major component of the tumor environment, promoting the establishment of a pro-invasive behavior. Such environment is supported by both tumor- and stromal-derived metabolites, particularly lactate. In prostate cancer (PCa), cancer-associated fibroblasts (CAFs) are major contributors of secreted lactate, able to impact on metabolic and transcriptional regulation in cancer cells. Here, we describe a mechanism by which CAF-secreted lactate promotes in PCa cells the expression of genes coding for the collagen family. Lactate-exploiting PCa cells rely on increased α-ketoglutarate (α-KG) which activates the α-KG-dependent collagen prolyl-4-hydroxylase (P4HA1) to support collagen hydroxylation. De novo synthetized collagen plays a signaling role by activating discoidin domain receptor 1 (DDR1), supporting stem-like and invasive features of PCa cells. Inhibition of lactate-induced collagen hydroxylation and DDR1 activation reduces the metastatic colonization of PCa cells. Overall, these results provide a new understanding of the link between collagen remodeling/signaling and the nutrient environment exploited by PCa.

**Keywords** CAFs; Lactate Metabolism; Collagen Hydroxylation; Collagen Signaling
**Subject Categories** Cancer; Cell Adhesion, Polarity & Cytoskeleton; Metabolism

## Introduction

During cancer progression, ECM plays both biophysical and biochemical roles to regulate/influence malignant cell behavior. The ECM is a meshwork of proteins that provides tumor cells the structural support for their spatial anchoring and adhesion within a tissue, together with biochemical signals that dynamically regulate tumor cells behavior during growth, survival, and invasion

(Cox, 2021). The major source of ECM within a tumor is the stroma compartment, particularly the CAFs population. So far, stromal ECM production and remodeling are considered a hallmark of cancer and have been associated with a poor prognosis (Provenzano et al, 2008; Ray et al, 2022). However, recent proteomics studies have also profiled 'the matrisome' (a collection of ECM and ECM-related proteins), of both primary tumors and metastases, highlighting that both tumor and stromal compartments can be defined as sources of ECM and ECM-affiliated molecules (e.g., glycoproteins, metalloproteases, integrins, etc.) (Socovich and Naba, 2019; Tian et al, 2019). Tumor cells sense the ECM through transmembrane receptors such as discoidin domain receptors (DDR1/2). DDRs, through downstream molecules (e.g., Signal Transducer and Activator of Transcription, STATs), coordinate tumor cell behavior including metastatic and motile ability (Dai et al, 2021).

The achievement of invasive traits by cancer cells has been associated with their interaction with stromal cells, likely forced to release exploitable metabolites. Lactic acid (LA) is one of the most abundant environmental metabolites within tumors and its levels correlate with a high likelihood of distant metastasis and worse outcome in cancer patients (Walenta et al, 2004). Tumor and tumor-associated glycolytic cell populations release it, and its enhanced transport and predominant usage of its carbon moieties are functional for metabolic, epigenetic and immune purposes (Faubert et al, 2017; Lopez Krol et al, 2022; Noe et al, 2021). In keeping, we demonstrated that CAF-released lactate supports PCa malignant phenotype by: (1) reshaping their metabolic behavior towards lactated-fueled OXPHOS (Fiaschi et al, 2012); (2) enhancing TCA cycle and lipid metabolism, (3) promoting histone acetylation and epigenetic regulation of pathways linked to an enhanced cancer aggressiveness (Comito et al, 2019; Ippolito et al, 2022a, 2022b; Ippolito et al, 2019). Cell-autonomous and non-cell-autonomous metabolic reprogramming in tumors can alter levels of metabolites likely dictating the activity of peculiar enzymes. For example, the TCA cycle intermediate α-KG can positively or negatively regulate the activity of the α-KG-dependent dioxygenases, including histone/DNA demethylases and hypoxia/collagen prolyl-hydroxylases (Abla et al, 2020), thereby serving as regulators of gene expression and impacting on ECM remodeling during tumor progression (D'Aniello et al, 2019; Elia et al, 2019).

[1]Department of Experimental and Clinical Biomedical Sciences, University of Florence, Viale Morgagni 50, 50134 Florence, Italy. [2]Institute of Oncology Research (IOR), Università della Svizzera Italiana (USI), 6500 Bellinzona, Switzerland. [3]Department of Minimally Invasive and Robotic Urologic Surgery and Kidney Transplantation, University of Florence, 50134 Florence, Italy. [4]These authors contributed equally: Luigi Ippolito, Assia Duatti, Elisa Giannoni, Paola Chiarugi. ✉E-mail: luigi.ippolito@unifi.it; paola.chiarugi@unifi.it

Here, we provide evidence for a new metabolic mechanism of ECM regulation driven by the upload of CAF-secreted lactate. Lactate-induced α-KG is exploited by P4H collagen prolyl-hydroxylase, impacting on collagen deposition. Moreover, we identified a type I collagen/DDR1/STAT3 axis as a signaling node required to promote a collagen-dependent achievement of pro-metastatic features in stromal lactate-reprogrammed PCa cells.

# Results

## LA sustains collagen synthesis and modification by controlling P4HA1 in prostate cancer cells

Nutrient abundance in the tumor environment influences cancer cell malignancy not exclusively for supporting intracellular cancer cells metabolism but also for the remodeling of ECM. High cell heterogeneity within solid tumors, e.g., breast and prostate leads to the accumulation of certain metabolites, including lactate, likely provided by different cellular sources, e.g., glycolytic CAFs and PCa cells themselves. As expected, extracellular levels of lactate from CAFs derived from human PCa specimens is higher than fibroblasts isolated from the benign hyperplasia tissues (Appendix Fig. S1a). With this regard, around 10 mM lactate has been found to be mostly produced in vitro by CAFs, with minor contribution from healthy fibroblasts (HPF) and PCa cells (Appendix Fig. S1b). Interestingly, lactate concentration in human xenografts reaches ~40 micromoles per gram of tissue (Appendix Fig. S1c), approximately corresponding to 40 mM. However, we chose 20 mM as the upper limit for our in vitro treatment, to better include the heterogenous contributions from the main cell populations. Furthermore, PCa cells have been demonstrated to be educated by CAFs contact to a respiratory behavior, exploiting the environmental lactate (Fiaschi et al, 2012), which is able to shape both metabolic and transcriptional dynamics in PCa cells (Ippolito et al, 2022a, 2022b). Along this line, we have evidenced that LA induces a peculiar ECM remodeling in PCa. Indeed, Gene Set Enrichment Analysis (GSEA) performed on the transcriptomic data of DU145 tumor cells exposed to exogenous LA for 48 h (Ippolito et al, 2022a, 2022b, Data ref: Ippolito et al, 2022a, 2022b) showed a positive correlation with gene sets associated with ECM remodeling and synthesis (Figs. 1A and EV1A). In keeping, a significant proportion of ECM-related genes (FC cutoff = 1, P value < 0.05) was significantly upregulated by LA exposure (Fig. 1B, red dots in the volcano plot; Dataset EV1 includes all the collagen genes regulated by the LA treatment). Moreover, we confirmed increased expression of collagen I (COL1A1, Col1) in PCa cells when exposed to conditioned media (CAF-CM) from lactate-rich CAFs compared to the lactate-poor fibroblasts (HPF-CM), obtained by healthy human prostatic tissues, as well as to LA, highlighting the role of environmental lactate, in sustaining cancer cell-derived Col1 synthesis (Fig. EV1B). Consistently, immunofluorescence microscopy revealed both higher intra- and extracellular Col1 levels and deposition in PCa cells exposed to the stromal lactate, which was blocked by the targeting of monocarboxylate transporter (MCT1)-mediated lactate uptake using the inhibitor AR-C155858 (Appendix Fig. S1D) or by RNAi (Figs. 1C and EV1C,D). Lactate drives a transcriptomic profile in cancer cells similar to that induced by hypoxia, called pseudohypoxia state (De Saedeleer et al,

2012). Accordingly, GSEA indicated that PCa cells are significantly enriched in hypoxic genes (Fig. EV1E). Interestingly, the expression of the α-KG-dependent prolyl-4-hydroxylase P4HA1, deputed to the proline hydroxylation needed for a proper collagen deposition, is induced by the hypoxia-inducible transcription factor (HIF-1) (Gilkes et al, 2013; Stegen et al, 2019). First, we found that both CAF and LA exposure significantly increase hydroxyproline content in DU145 cells, as readout of P4H activity. The addition of cell-permeable α-KG rescues the reduced hydroxyproline levels observed when the MCT1-mediated LA entry is inhibited (Fig. 1E). In keeping with hypoxic signature, we also observed that P4HA1 expression is increased in PCa cells in response to CAF/LA conditioning (Figs. 1E and EV1F). To sustain the role of lactate in CAF-CM, we boiled it to eliminate the contribution of protein factors. Interestingly, both inhibition and genetic depletion of MCT1 significantly decrease CAF-CM and LA-induced expression in PCa cells (Figs. 1E and EV1G), and boiled CAF-CM was still sufficient to induce P4HA1 expression. To note, a higher affinity for α-KG is relative to P4HA1 compared to other α-KG-dependent prolyl-hydroxylases (Hirsilä et al, 2003). Accordingly, we observed that CAF-CM or LA-treated DU145 cells exhibit higher availability of α-KG, but the genetic depletion of P4HA1 (Fig. EV2A) mostly leads to a further increase in α-KG levels (Fig. 1F), as well as to a reduction of hydroxyproline content (Fig. EV2B), indicating that P4HA1 tunes LA-induced α-KG to support its function. In keeping, we observed that P4HA1 silencing negatively impacts on LA-promoted collagen deposition in PCa cells, as shown by immuno-fluorescence of extracellular Col1 (Fig. EV2C,D). The clinical relevance of these findings was also confirmed by in silico analysis. Indeed, we observed a significant association between *P4HA1* and *MCT1* expression in prostate cancer patients' specimens (Fig. 1G), suggesting that P4HA1 can have a role in lactate-rich prostate carcinomas. To further confirm the association between P4HA1 expression and prostate cancer progression, we analyzed P4HA1 protein levels in human prostate cancer tissues using tissue microarrays (TMA). P4HA1-positive staining was significantly enriched in high-Gleason PCa and in metastatic PCa tissues (Figs. 1H and EV2E). These data highlight the crucial role of LA as a metabolic regulator of collagen remodeling in prostate cancer.

## LA-dependent P4HA1 is crucial for the invasiveness of prostate cancer cells

Deposition, remodeling, and signaling of the extracellular matrix facilitate tumor growth and metastasis. We examined whether LA-induced P4HA1 could help to potentiate the invasive ability of CAF-exposed prostate cancer cells. Silencing of P4HA1 impaired the invasive ability of PCa cells exposed to either CAF-CM or LA (Figs. 2A and EV3A). Also, P4HA inhibitor ethyl-3,4-dihydrox-ybenzoic acid (DHB) administration is able to impair both CAF and LA stimuli for PCa cell motility (Fig. EV3B–D). Together with invasion, collagen remodeling enhances cancer cell extravasation and premetastatic niche formation (Elia et al, 2019; Han et al, 2016). We showed that CAF-CM and LA-stimulated PCa cells increase the transendothelial migration ability of cancer cells, and this increase was impaired by P4HA1 genetic depletion (Fig. 2B). Importantly, we observed that knockdown of P4HA1 significantly decreased tumor spheres number and size in CAF/LA-exposed PCa cells (Figs. 2C and EV3E). Finally, we investigated whether

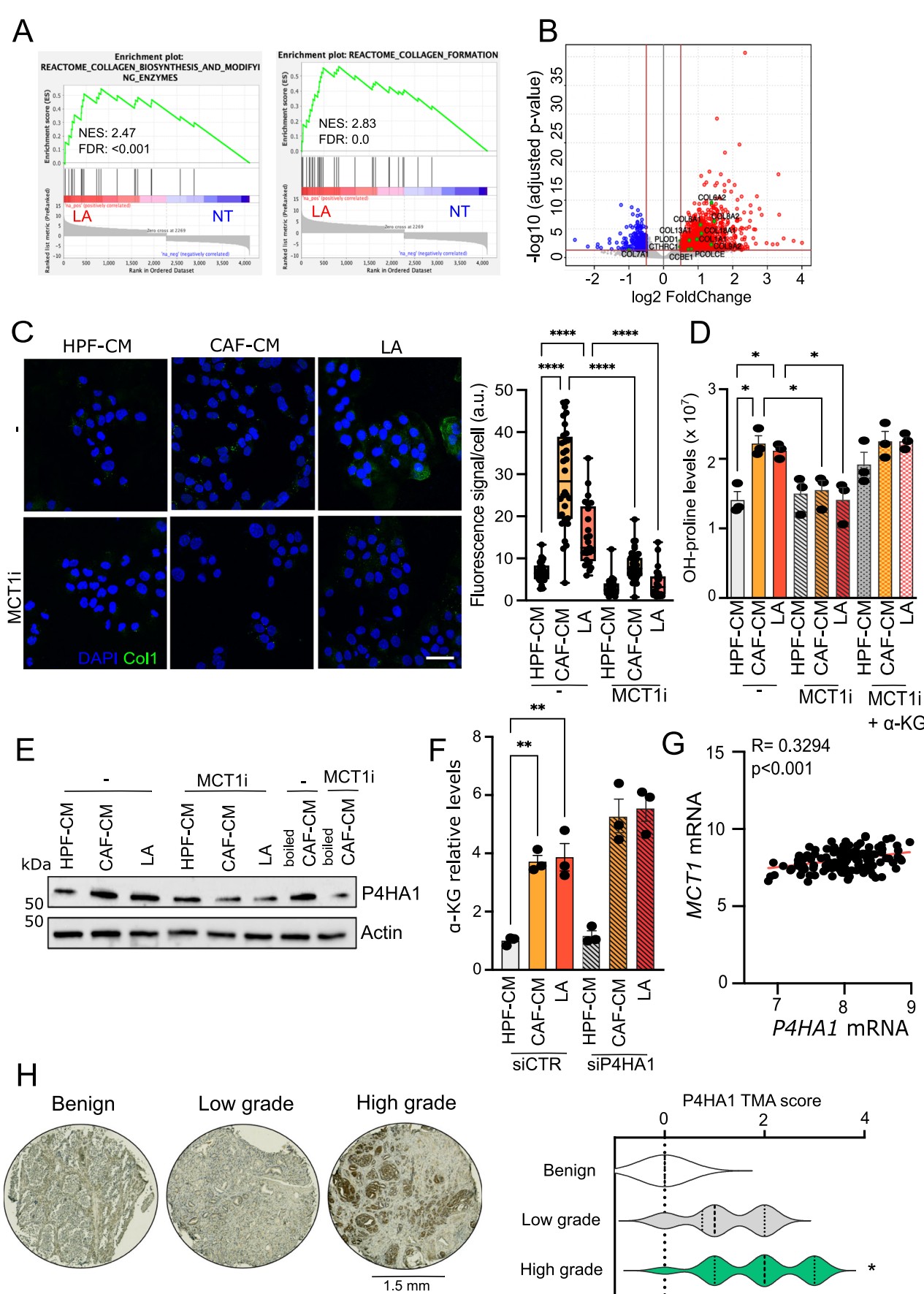

◀  **Figure 1.   Environmental lactate enhances collagen synthesis by sustaining P4HA1 in PCa cells.**

(A) Enrichment plots of the Reactome Collagen Biosynthesis and Modifying Enzymes and Collagen Formation pathways showing a positive association between these MsigDb datasets and the LA-exposed DU145 gene expression profile. NES normalized enrichment score. (B) Differential gene expression analysis between LA-exposed (20 mM for 48 h) and untreated (serum-starved) DU145 cells. The gray dots represent all the considered genes. Red and blue dots represent, respectively, the upregulated and downregulated genes in the over a 0.5 |log2FC| considering a P value cutoff equal to 0.05. Collagen genes are highlighted in green. (C) DU145 were exposed to HPF-, CAF-CM or LA ± AR-C155858 (MCT1i, 40 μM) and stained for collagen I (green: intracellular collagen I; blue: DAPI, nuclei; representative pictures were shown, scale bar, 10 μm. Box plot showing the quantification of fluorescence signal per cell: centerlines show the medians; box limits indicate the 25th and 75th percentiles; and whiskers extend to the minimum and maximum. n = 25-42 cells from three biological replicates. (D) Hydroxyproline content from DU145 cells treated as indicated ± MCT1i or cell-permeable α-KG (2 mM). Each dot represents a biological replicate. (E) Representative western blot analysis of P4HA1 in DU145 cells, treated as indicated, ± MCT1i. Beta-actin was used as a loading control. (F) GC-MS analysis of α-KG levels in DU145 silenced for P4HA1 (siP4HA1) and for a siRNA control (siCTR). (G) Correlation analysis between P4HA1 and MCT1 expression in primary tumor specimens of Taylor dataset (n = 183). Each dot corresponds to an individual specimen. Pearson correlation r. (H) P4HA1 expression in representative tissue cores from TMA (n = 63; magnification ×20), including benign and tumor tissue samples with low (< 3) and high Gleason grade (≥3). Data information: (A, B) data are represented as means of n = 3 biological replicates and P values were calculated by DESeq2 with Benjamini–Hochberg multiple test correction (B). GSEA normalized enrichment scores (NES) and FDR values are indicated in (A). Bar graphs in (C, D, F) represent means ± SEM, n = 3 biological replicates. Significance was determined using one-way ANOVA, followed by Tukey's multiple comparisons test (*P < 0.05; **P < 0.01; ***P < 0.001; ****P < 0.0001). Source data are available online for this figure.

CAF/LA-induced P4HA1 expression contributes to cancer cell colonizing behavior. CAF-CM and LA pre-treated control (siCTR) and P4HA1-silenced PCa cells were labeled with CellTracker Green or Red dyes, respectively, 1:1 mixed and injected *via* tail vein into SCID mice. Imaging of the lungs at five hours after injection revealed that a reduced number of P4HA1-silenced PCa cells were retained in the lungs, compared to their control cells exposed to CAF-CM and LA (Figs. 2D and EV3F). These data suggest that P4HA1 is required for an efficient retention in the lung, the first key step in metastatic colonization.

## LA-guided type I collagen promotes invasive traits in PCa cells *via* DDR1

DDR1 is a non-integrin receptor that binds fibrillar collagens, and its activation has been found in metastatic tumor populations (Gao et al, 2016; Su et al, 2022). We hypothesized the involvement of DDR1 since its expression is higher in PCa metastatic tissues, compared to the primary tumors, and is positively correlated with P4HA1 (Figs. EV4A and 3A). Interestingly, we observed that stromal LA conditioning promotes DDR1 expression (Figs. 3B and EV4B), a trait that was impaired by MCT1 inhibition. Intriguingly, emerging evidence showed that DDR1 is linked to the metastatic traits enrichment, responsible for multi-organ metastasis (Gao et al, 2016). Therefore, we found that the invasive ability of DDR1-silenced cells (Fig. EV4C,D) was impaired upon CAF/LA stimulation (Figs. 3C and EV4E). Similarly, N-isopropyl-4-methyl-3-(2-(pyrazolo-[1,5-a]pyrimidin-6-yl)ethynyl) 7rh benzamide (7rh), a potent DDR1 inhibitor (DDRi), effectively blocks LA-driven invasion (Fig. EV4F). Interestingly, DDR1 silencing strongly reduces tumor sphere formation sustained by CAF and LA conditioning in PCa cells (Figs. 3D and EV4G), an effect accomplished also by 7rh administration (Fig. EV4I,J). In agreement, DDR1 silencing and inhibition significantly suppress the expression of key stemness markers, such as SOX2 (Figs. 3E and EV4H) and ALDH1A1 (Fig. 3F), corroborating our idea that DDR1 is useful to maintain the stemness phenotype in LA-exploiting tumor cells. Again, the ability to intra/extravasate sustained by LA was dampened in PCa cells exhibiting DDR1 knockdown or pharmacological blockade (Figs. 3G and EV4K). Finally, DDR1 expression was strongly enriched in high Gleason-graded PCa patient samples compared to the low-graded ones, as shown by TMA cores (Fig. 3H). These data suggest that PCa cells displaying lactate-driven metastatic

potential exhibit, in addition to the collagen remodeling, a collagen sensing and signaling orchestrated by DDR1.

## Tumor cell-derived Col1 is functional for DDR1 activation

Conflicting insights have emerged to address the role of collagen in supporting or delaying tumor progression (De Martino and Bravo-Cordero, 2023). We hypothesized a peculiar role of tumor cell-derived collagen induced by stromal lactate, specifically acting as a signaling molecule *via* DDR1. Firstly, we observed a positive correlation between DDR1 and COL1A1 expression in a cohort of PCa patients (Fig. 4A). To further support the specific role of the epithelial collagen in activating DDR1, we allowed CAFs or LA-stimulated PCa cells to deposit their own matrices (Fig. EV5A). Upon decellularization, cancer cells were seeded on these matrices and analyzed for DDR1 expression. We found that matrix deposited by cancer cells under LA stimulation, but not matrix deposited by CAFs, is supportive in inducing DDR1 activation by phosphorylation in LA-reprogrammed PCa cells (Fig. 4B), indicating that only cancer cell secreted matrix is critical in activating DDR1. Recent studies have suggested that cancer cells, including PCa cells, specifically produce unique Col1 homotrimer compared to normal type I collagen heterotrimer (α1/α2/α1) produced by fibroblasts (Chen et al, 2022; Makareeva et al, 2010). In agreement with this finding, in LA-treated PCa cells we observed the upregulate of *Col1a1* chain only (Fig. EV5B), while human CAFs actively express both a1 and a2 chains (*Col1a2*), suggesting that differential collagen oligomerization in stromal and cancer cells may be critical for DDR1 activation. Then, we used COL1A1-silenced PCa cells (Fig. EV5C) and we observed that CAF/LA-induced invasiveness (Fig. 4C), as well as the prostasphere formation (Figs. 4D and EV5D) were strongly impaired. These data suggest that cancer cell-derived collagen induced by environmental lactate sustains invasive traits in a para- or autocrine manner.

## LA-dependent DDR1/STAT3 pathway regulates type I collagen expression and function in PCa cells

DDR1 signaling cascade activation in metastatic cells is carried out through different downstream molecules, including STAT3 (Gao et al, 2016). In silico analysis of PCa primary and metastatic tissues

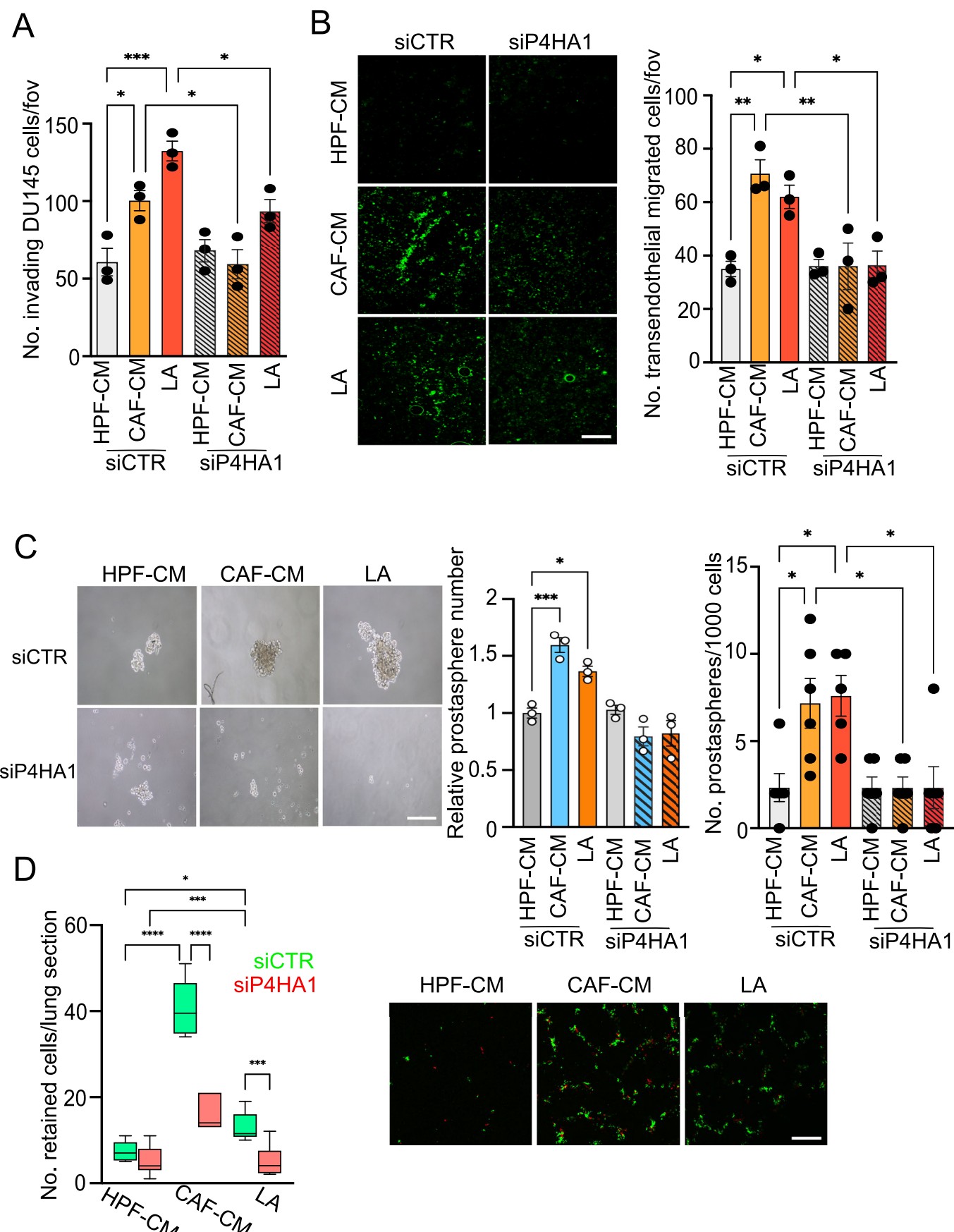

**Figure 2. LA-induced P4HA1 promotes invasive features in PCa cells.**

DU145 were silenced using the non-targeting control (siCTR) or P4HA1-specific pool of siRNAs and subsequently treated as indicated. (A) Invasion assay performed on DU145 cells treated as indicated. (B) Fluorescently green-labeled DU145 cells, silenced for P4HA1 and treated as indicated, were allowed to migrate through a HUVEC confluent monolayer. Representative pictures of transendothelial migrating green-fluorescent tumor cells are shown, scale bar: 100 μm. (C) Prostaspheres formation assay of DU145 cells treated as indicated. The number and relative size of spheres at day 10 were quantified and representative pictures are shown. Scale bar: 100 μm. (D) Cells were labeled with CellTracker dyes (green for siCTR and red for siP4HA1) before injecting a mixture of siCTR and siP4HA1 transfected cells in a 1:1 ratio into the tail vein of SCID mice. The lungs were recovered at the time indicated and imaged to analyze the number of fluorescent cells that colonize the lungs. Data shown are means for number of tumor cells per lung section; $n = 6$–12 sections. Representative images are shown; scale bar: 200 μm. Quantification of green and red fluorescent cells per lung section was reported as box plot: centerlines show the medians; box limits indicate the 25th and 75th percentiles; and whiskers extend to the minimum and maximum. Data information: bar graphs in (A–D) represent means ± SEM, $n = 3$ biological replicates (A–C), for prostasphere number (C) $n = 1$–2 technical replicates from each biological experiment, $n = 4$ mice/group (D). Significance was determined using one-way ANOVA, followed by Tukey's multiple comparisons test (*$P < 0.05$; **$P < 0.01$; ***$P < 0.001$; ****$P < 0.0001$). Source data are available online for this figure.

showed a positive correlation between DDR1 and STAT3 expression (Fig. 5A). Thus, we assessed this interplay and we found that PCa cells exposed to CAF-CM or LA show an increased phosphorylation of STAT3 in tyrosine 705 (p-STAT3), while DDR1 knockdown negatively affects STAT3 activation in these cells (Fig. 5B). Accordingly, DDR1 pharmacological inactivation by 7rh administration strongly dampens p-STAT3 levels in PCa cells (Appendix Fig. S2A,B). Moreover, STAT3 phosphorylation (even tested in CAF-CM boiled samples to eliminate protein factors, e.g., cytokines) is negatively impacted by (i) P4HA1 genetic and pharmacological impairment (Appendix Fig. S2C,D) and (ii) Col1 silencing (Appendix Fig. S2E) in CAF/LA-treated cells. Importantly, western blot analysis showed that LA-conditioned PCa cells, upregulating DDR1 expression, when seeded on LA-induced tumor decellularized ECM, exhibit both higher DDR1 and STAT3 phosphorylation/activation (Figs. 4B and 5C). Therefore, the LA-dependent P4HA1/Col1/DDR1 axis is critical for STAT3 activation. Then we examined whether STAT3 activation phenocopies the cancer cell-derived collagen effects. We observed that STAT3 silencing negatively impacts on the CAF/LA enhanced invasiveness and prostaspheres formation (Fig. 5D,E). In parallel, STAT3 targeting by the selective STAT3 inhibitor Stattic (STAT3i) leads to similar results (Appendix Fig. S3A–C), thus demonstrating a key role of STAT3 activation in controlling the invasive and stem-like features of LA-rewired PCa cells. Next, we investigated STAT3 as a transcription factor potentially maintaining the expression of DDR1 and Col1. Interestingly, the analysis of the prediction of binding sites revealed that both DDR1 and Col1 have a predicted STAT3 binding site at the promoter region (JASPAR Predicted Transcription Factor Targets, Appendix Fig. S3D). To note, immunofluorescence and immunoblot analyses showed that LA-driven DDR1 and Col1 expression are significantly reduced in STAT3-silenced or STAT3-inactivated PCa cells exposed to stromal LA (Fig. 5F,G; Appendix Fig. S3E). These data reveal a lactate-dependent loop where binding of DDR1 to type I collagen triggers STAT3 activation to regulate and maintain Col1 expression.

### Lactate promotes the metastatic burden of prostate cancer cells in vivo through collagen-dependent DDR1 activation

To finally examine the impact of lactate-sustained P4HA1-DDR1 axis in the metastatic behavior of PCa cells in vivo, we stably silenced PCa cells for P4HA1 or DDR1 (Appendix Fig. S4A), then

establishing tumor xenografts in SCID mice by subcutaneously (s.c.) co-injecting scrambled (Scr) or P4HA1- or DDR1-silenced cells with HPFs, CAFs or alone, and then exposing the xenografted mice to a daily i.p. treatment with lactate or vehicle. IHC examination revealed higher expression of DDR1 and P4HA1 in scrambled xenograft tumors treated with lactate or co-injected with CAF compared to the HPF-related controls. Low expression of both P4HA1 and DDR1 was confirmed in knockdown xenograft tumors (Fig. 6A,B). Analysis of the tumor weight 60 days after s.c.injection revealed a higher weight of the tumor masses derived from CAF and lactate scrambled tumor xenografts compared to both DDR1 and P4HA1-silenced counterparts, although not statistically significant (Fig. 6C). However, Col1 deposition is highly increased in CAF and lactate-experiencing tumor masses (Appendix Fig. S4B). Moreover, both hematoxylin/eosin (HE) and human pancytokeratin staining of the lungs showed the presence of lung metastatic depots, that are higher in the animals engrafting scrambled tumor cells experiencing in vivo CAFs or lactate conditioning, than the shP4HA1 and shDDR1-derived specimens (Fig. 6D; Appendix Fig. S4C,D), suggesting an important involvement of these two players in lactate-sustained tumor lung colonization. Finally, survival analysis indicated that patients with higher P4HA1 and Col1a1 expression were significantly associated with worse disease-free survival, supporting the fact that P4HA1-induced collagen I remodeling is an important feature in highly aggressive PCa (Fig. 6F,G). Collectively, these results suggest that the lactate-sustained P4HA1/DDR1 axis contributes to the cancer cell colonization and the formation of metastasis in vivo.

## Discussion

The study presented here identifies at least three novel findings: (i) lactate, massively secreted by CAFs, is a trigger for the tumor-derived matrisome remodeling through the activation of the collagen prolyl-hydroxylase P4HA1 and de novo collagen I synthesis; (ii) the autocrine secretion of collagen by cancer cells is a key determinant of their invasive phenotype; (iii) the P4HA1-DDR1-STAT3 axis is the molecular pathway driving such pro-metastatic cues. Overall, our work describes a new mechanism characterizing a highly invasive cancer cells population experiencing a lactate-rich environment, i.e., primary prostate tumor tissue, leading to the activation acquisition of a pro-metastatic program by PCa cells exhibiting a particular matrisome signature upon CAF-derived lactate conditioning.

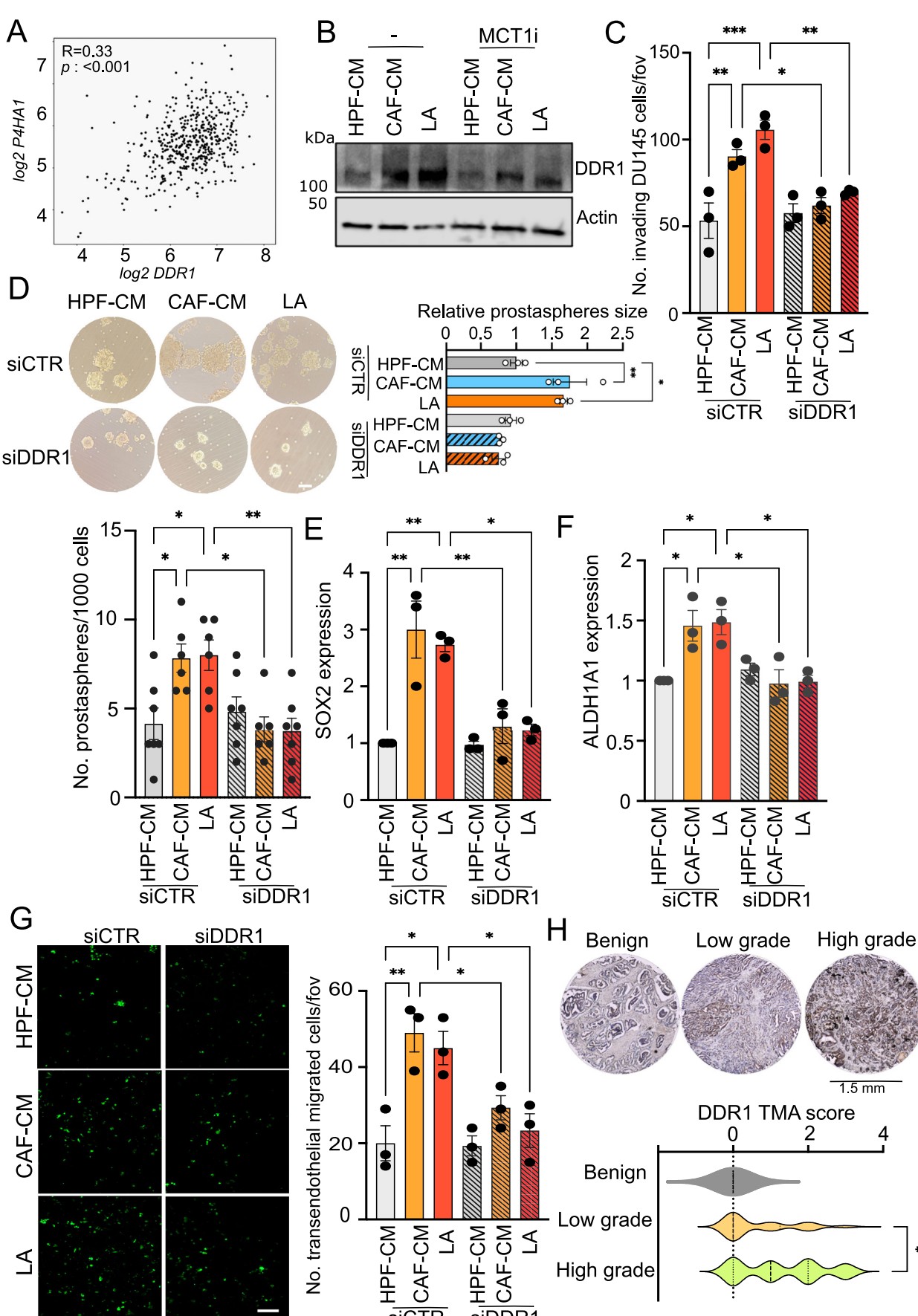

◄ **Figure 3. DDR1 activation supports the achievement of LA-induced stem-like and invasive features in PCa cells.**

(A) Correlation analysis between P4HA1 and DDR1 expression in primary tumor specimens of TCGA-PRAD dataset ($n = 497$). Each dot corresponds to an individual specimen. Pearson correlation $r$. (B) Representative western blot analysis of DDR1 in DU145 cells, treated as indicated, ± MCT1i. Beta-actin was used as a loading control. (C–E) DU145 cells were silenced using the non-targeting control (siCTR) or DDR1-specific pool of siRNAs and subsequently treated as indicated. Invasion assay (C), prostaspheres formation assay (representative pictures at day 10 are shown, scale bar: 100 μm) (D) and qRT-PCR for SOX2 and ALDH1A1 analysis (E, F) in DDR1-silenced DU145 cells using siCTR-treated cells as comparators. (G) Fluorescently labeled DU145 cells, silenced for DDR1 and treated as indicated, were allowed to migrate through a HUVEC confluent monolayer. Representative pictures of transendothelial migrating green-fluorescent tumor cells are shown, scale bar: 100 μm. (H) DDR1 expression in representative tissue cores from TMA ($n = 63$; magnification, ×20) including benign and tumor tissue samples with low (< 3) and high Gleason grade (≥3). Violin plots of DDR1 histoscores in tissue samples compared by Gleason grade. Data information: bar graphs in (C–H) represent means ± SEM; (C–G) $n = 3$ biological replicates, for prostasphere number (D) $n = 1$–2 technical replicates from $n = 3$ biological replicates, for (H) means of histoscores. Significance was determined using one-way ANOVA, followed by Tukey's multiple comparisons test (*$P < 0.05$; **$P < 0.01$; ***$P < 0.001$; ****$P < 0.0001$). Source data are available online for this figure.

We identified the member of the collagen prolyl-hydroxylase P4HA1 to be activated by lactate conditioning in PCa cells. P4HA1 and other ECM-related enzymes have been associated to higher tumor aggressiveness in different cancer models and its expression is transcriptionally controlled by HIF-1 (Stegen et al, 2019; Xiong et al, 2018). Lactate entry and exploitation activate HIF-1 within tumor cells (Ippolito et al, 2019). Also, the availability of α-KG content is crucial for the activity of certain α-KG-dependent enzymes including P4HA1. Indeed, α-KG levels induced by lung-derived pyruvate support the 3D growth of metastasizing breast cancer cells, by driving collagen hydroxylation and ECM remodeling (Elia et al, 2019). Also, α-KG consumption by P4HA1 may regulate a HIF-1-dependent tumor stemness and drug resistance in breast cancer cells (Xiong et al, 2018). Similarly, our data support a P4HA1-induced ECM remodeling driven by stromal lactate. This may represent also a mechanism tuning cellular α-KG levels, that, if too increased, can abrogate the (pseudo)hypoxic state of PCa cells, and consequently the HIF-1-dependent response, known to be critical for PCa behavior (Fiaschi et al, 2012).

ECM remodeling events can support the invasion of tumor cells at different steps of metastatic cascade. Interestingly, circulating tumor cells (CTCs) upregulate expression of common stroma-derived ECM proteins, such as collagens (Ting et al, 2014). Together with a peculiar hypoxic signature driving P4HA1 expression in CTCs clusters (Donato et al, 2020), CTC-secreted ECM may increase CTCs survival in the bloodstream, protect them from the immune clearance and promote their clustering through collagen embedding or platelet recruitment (Xiong et al, 2020), thereby enhancing the efficacy of their metastatic engraftment. Alongside these findings, we observed that lactate-promoted collagen remodeling in PCa cells is mandatory in sustaining the survival of cells within the circulation in the lung retention assay, as wells as cell clustering and 3D sphere growth in vitro.

Remodeling of collagen fibers as well as the ECM deposition and stiffness have been described as important actors in PCa progression, i.e., the metastatic prostate cancer (Caley et al, 2016). In this context, it is likely that tumor-associated stroma is the source of such collagens, having both tumor-promoting and tumor-restricting activity depending on the context (Tian et al, 2019). Therefore, a collagen-poor stroma is responsible for an undifferentiated and invasive phenotype which accelerates pancreatic cancer (PDAC) progression and decreases overall survival, suggesting that stromal collagens may inhibit tumor progression (Chen et al, 2021). In parallel, ECM components produced by the tumor cells, rather than from the fibroblasts, are likely associated with poor patient survival (Tian et al, 2020). Our study fits with

PDAC scenario, highlighting that the tumor-derived collagen is pro-invasive for PCa cells. A recent study has shown that, in contrast to normal type I collagen (Col1) heterotrimer (α1/α2/α1) produced by fibroblasts, pancreatic cancer cells specifically produce unique Col1 homotrimer (α1/α1/α1) because of an epigenetic suppression of the α2 chain gene (Chen et al, 2022). A seminal paper demonstrated that homotrimeric collagen appears to be produced by cancer cells only, including PCa cell lines (Makareeva et al, 2010). Alongside, we speculated that lactate may stimulate the deposition of collagen I homotrimers in PCa cells. As these collagen structures are resistant to collagen-degrading enzymes, PCa cells may utilize them likely to establish roadways for invasion and metastasis.

Lactate has been demonstrated to potentiate both the invasiveness of tumor cells in vitro and in vivo (Ippolito et al, 2022a, 2022b). In this regard, we identified a cancer cell-derived collagen receptor DDR1 as a sensor of tumor-secreted collagen I, which enhances stem-like and metastatic behavior in vitro and in vivo. Recently, several roles of DDR1 in the metastatic process have been uncovered, ranging from the support for the colonization (Dai et al, 2021), to regulation of both dormant and metastatic reactivated cancer cell state (Di Martino et al, 2022; Gao et al, 2016), to immune escaping (Sun et al, 2021), accordingly to the architecture of the ECM deposited at the primary/metastatic tissue. In such scenario, it is important to highlight that solid tumors in in vivo context may simultaneously experience both collagen-promoting and collagen-restricting forces, and DDR1 can sense different collagens according to the distinct microenvironments at the local and metastatic sites. However, there are still confounding findings on which type of collagens is active on DDR1, even in the same tumor type. In PDAC, two different studies have shown opposite conclusions, demonstrating that stromal-cleaved collagen triggers DDR1 via NF-κB signaling, thereby enhancing mitochondrial dynamics and tumor growth (Su et al, 2022), while the homotrimeric (i.e., intact) collagen activates DDR1/FAK/ERK in a cell-autonomous manner (Chen et al, 2022). Also, the orientation of collagen fibers is crucial for DDR1 activation, for example wavy rather than aligned type III collagen is sensed by DDR1-expressing dormant tumor cell (Di Martino et al, 2022). Molecularly, we found that lactate regulates the activation of P4HA1/COL1/DDR1 axis and that DDR1 signaling via STAT3 maintains both DDR1 and COL1A1 expression. In keeping, JAK–STAT activation has been demonstrated to be specifically required for the transition to a stem-like, EMT-associated, therapy-resistant state in PCa (Deng et al, 2022). Inhibition of STAT3 by Napabucasin, a Phase II/III FDA-approved drug, significantly abrogated the capacity of

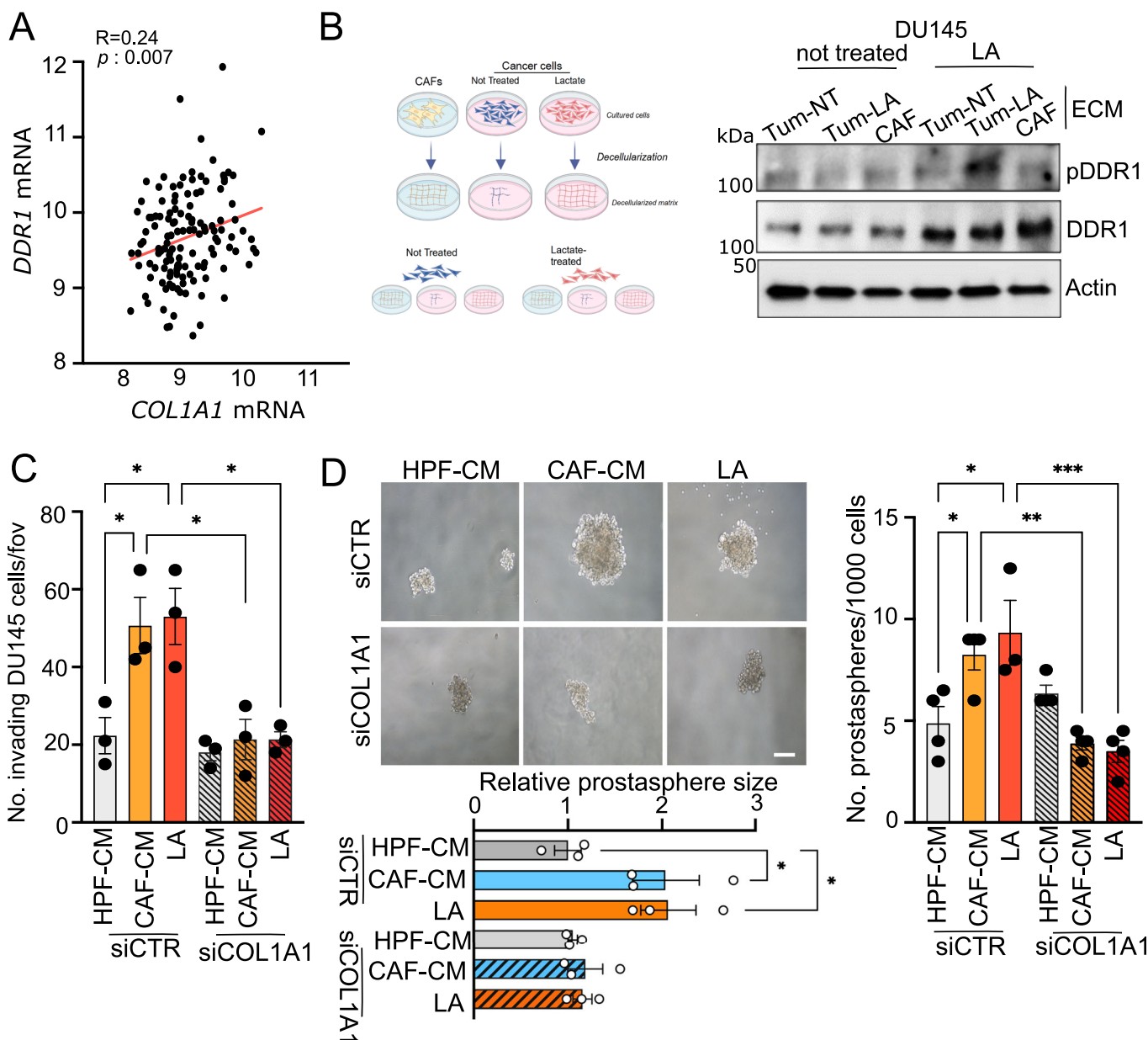

Figure 4. LA-induced collagen I is crucial for the invasiveness of PCa cells.

(A) Correlation analysis between COL1A1 and DDR1 expression in primary tumor specimens of Taylor dataset ($n = 183$). Each dot corresponds to an individual specimen. Pearson correlation $r$. (B) Schematic explanation of the experimental setting: the tumor cell DDR1 activation (not-treated or LA-treated cancer cells) to three decellularized matrices (produced by CAFs, not-treated or LA-administered tumor cells, respectively) was evaluated by western blot. Representative western blot analysis of p-DDR1(Tyr702), DDR1 in DU145 cells, untreated or not with LA, allowed to grow onto decellularized matrices (ECM) derived from not-treated or LA-conditioned tumor cells (Tum-NT; Tum-LA) or CAFs. (C) Invasion assay performed on DU145 cells treated as indicated. (D) Prostaspheres formation assay of DU145 cells silenced for COL1A1, treated as indicated. The number and relative size of spheres at day 10 were quantified and representative pictures are shown. Scale bar: 100 μm. Data information: bar graphs in (C, D) represent means ± SEM; (C, D) $n = 3$ biological replicates; for prostasphere number (D) $n = 1$-2 technical replicates, $n = 4$ biological replicates. Significance was determined using one-way ANOVA, followed by Tukey's multiple comparisons test (C, D) (*$P < 0.05$; **$P < 0.01$; ***$P < 0.001$). Source data are available online for this figure.

DDR1-expressing cancer cells to form lung metastases from bladder cancer (Lee et al, 2019). Interestingly, DDR1 can even translocate to the nucleus (Chiusa et al, 2019), suggesting that additional DDR1-dependent mechanisms may regulate ECM gene expression.

Overall, in our study we documented that lactate acts as a prominent source for supporting collagen-producing PCa

efficiently metastasizing cells. Hence, targeting of lactate exploitation by PCa cells, through inhibition of MCTs transporters, may associate to the impairment of motility and invasive abilities (Ippolito et al, 2019; Tasdogan et al, 2020), to the suppression of stem-like phenotype, anchorage independence and 3D overgrowth, key determinants for successful metastatic cells. Moreover, while it

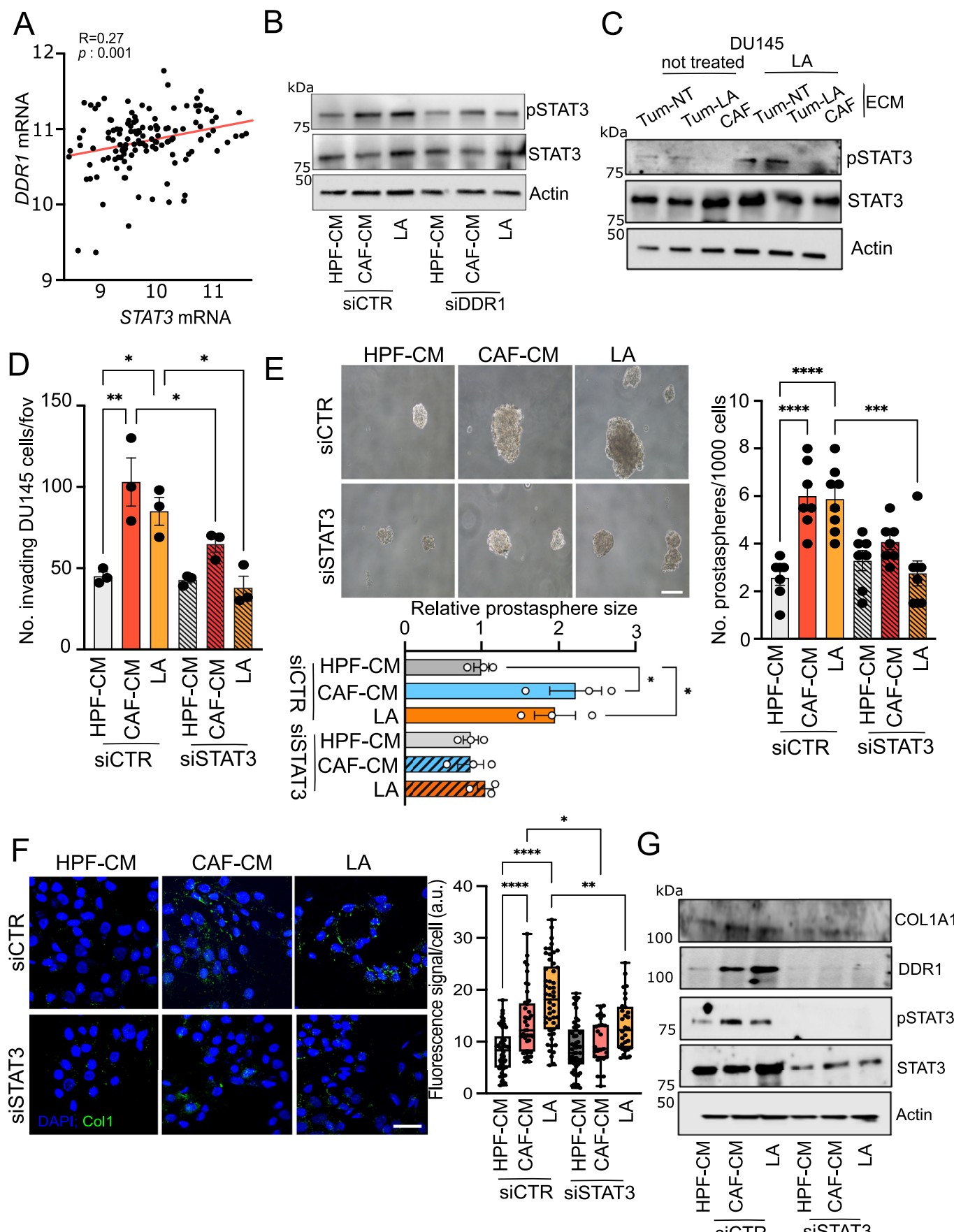

**Figure 5. STAT3 activation supports the DDR1 signaling in LA-reprogrammed PCa cells.**

(A) Correlation analysis between STAT3 and DDR1 expression in primary tumor specimens of Taylor dataset ($n = 183$). Each dot corresponds to an individual specimen. Pearson correlation $r$. (B) Representative western blot analysis of p-STAT3(Tyr705), STAT3 in DU145 silenced for DDR1, treated as indicated. Beta-actin was used as a loading control. (C) Representative western blot analysis of p-STAT3(Tyr705), STAT3 in DU145, pre-treated with LA, allowed to grow onto decellularized matrices (ECM) derived from not-treated or LA-conditioned tumor cells (Tum-NT; Tum-LA) or CAFs. (D) Invasion assay performed on DU145 cells treated as indicated. (E) Prostaspheres formation assay of DU145 cells silenced for STAT3, treated as indicated. The number and relative size of spheres at day 10 were quantified and representative pictures are shown. Scale bar: 100 μm. (F) Representative pictures of collagen I (green) from immunofluorescence analysis of DU145 silenced for STAT3, treated as indicated. Nuclei (blue) were stained with DAPI Scale bar: 10 μm. Box plot showing the quantification of fluorescence signal per cell: centerlines show the medians; box limits indicate the 25th and 75th percentiles; and whiskers extend to the minimum and maximum. $n = 27$–53 cells from three biological replicates. (G) Representative western blot analysis of COL1A1, DDR1, p-STAT3(Tyr705), STAT3 in DU145 silenced for STAT3, treated as indicated. Beta-actin was used as a loading control. Data information: bar graphs represent means ± SEM; (D–F) $n = 3$ biological replicates; for prostasphere number (E) $n = 1$–2 technical replicates from four biological replicates. Significance was determined using one-way ANOVA, followed by Tukey's multiple comparisons test ($*P < 0.05$; $**P < 0.01$; $***P < 0.001$; $****P < 0.0001$). Source data are available online for this figure.

is conceivable that collagen itself is not an ideal druggable target, a promising pharmacological approach may also be the small-molecule compounds that inhibit collagen remodeling or collagen-receptor interaction that have been characterized and developed recently (i.e., collagen prolyl-hydroxylases and DDRs inhibitors) (Wang et al, 2020; Zhavoronkov et al, 2019).

# Methods

## Cell models

Human prostate cancer cells (DU145, RRID:CVCL_0105; PC3, RRID:CVCL_0035; 22RV1, RRID:CVCL_1045) and HUVEC were obtained from ATCC. Human prostate fibroblasts [healthy prostate fibroblasts (HPFs) isolated from benign prostate hyperplasia and CAFs] were isolated from surgical explants after patient informed consent, according to the Ethics Committee of the Azienda Ospedaliera Universitaria Careggi (Florence, Italy). An informed consent was obtained from all subjects and that the experiments conformed to the principles set out in the WMA Declaration of Helsinki and the Department of Health and Human Services Belmont Report. CAFs specimens were collected and selected from patients according to the Gleason score, PSA levels and positivity for specific marker (e.g., FAP). All cells were maintained in DMEM (#ECB7501L; Euroclone) supplemented with 10% FBS (#ECB4004L; Euroclone), 2 mmol/L L-glutamine and 1% penicillin/streptomycin, except for HUVEC cultured in EGM-2 (Lonza, #CC-3162). All cell lines were maintained at 37 °C and 5% CO$_2$ and were routinely tested for Mycoplasma contamination with the MycoAlert Mycoplasma Detection kit (#LT07–318; Lonza).

## Conditioned media from fibroblasts

HPFs and CAFs were grown to subconfluence and treated for 48 h with serum-free medium to obtain the corresponding conditioned media (CM), which are filtered and used fresh or stored for further analysis. When indicated, CM was boiled for 15 min at 95 °C, to eliminate protein factors.

## Cell treatments and transfection

Cell treatments were performed in HPF- or CAF-CM or in serum-free culture medium supplemented with 20 mmol/L lactic acid (Merck, #L1750) for 48 h. Unless specified otherwise, all reagents were obtained

from Merck Millipore. The following inhibitors were used in this study: 40 μmol/L MCT1 inhibitor AR-C155858 (#4960; Tocris BioScience), 20 μmol/L 3,4-dihydroxybenzoate DHB (Merck, #E24859), 500 nmol/L 7rh (Merck, # SML1832), 2,5 μmol/L Stattic (Merck, #573099). Cell-permeable α-ketoglutarate (dimethyl 2-oxoglutarate) was purchased from Sigma-Aldrich and used at the concentration of 2 mM. Cells were transfected with a pool of siRNA targeting human prolyl-4-hydroxylase A1 (P4HA1, EHU081741), Discoidin Domain receptor 1 (DDR1, EHU113231), collagen1a1 (COL1A1, EHU153561) and STAT3 (EHU122051) using Lipofectamine RNAiMAX transfection reagent (#13778150; Thermo Fisher), according to the manufacturer's instructions. Lentiviral vectors containing short hairpin for P4HA1 and DDR1 were obtained from shRNA Mission (Sigma). To obtain viral particles, HEK293T packaging cells were transfected with 8 μg of plasmid by calcium phosphate method (Promega, #E1200). Empty and scramble vectors were used to obtain mock and scramble-infective viral particles. Two days after transfection, supernatants were centrifuged for 10 min at $600 \times g$ and filtered through a 0.45-μm-pore filter. For transduction, cells were seeded at $1 \times 10^5$ and incubated overnight with viral supernatants in the presence of 8 μg/mL polybrene (Sigma). Forty-eight hours post-infection, cell populations were incubated in a medium containing the puromycin (1 μg/mL) for 2 additional weeks.

## Lactate measurement

Lactate concentrations in cell supernatants or in tissues were determined enzymatically using L-Lactate assay kit (#MAK329, Merck), following the manufacturer's instructions.

## RNA extraction and qRT-PCR

Total RNA was extracted using RNeasy Kit (#74104; Qiagen), and cDNA synthesis was performed using iScript cDNA Synthesis Kit (#1708891; Bio-Rad). qRT-PCR was carried out by CFX96 Touch Real-Time PCR Detection System (Bio-Rad) using TaqMan assays (Life Technologies). The following probes were used in this study: DDR1 (Hs01058430_m1), COL1A1 (Hs00164004_m1), COL1A2 (Hs01028956_m1), SOX2 (Hs04234836_s1), ALDH1A1 (Hs00946916_m1). Data were normalized on HPRT1 (#Hs02800695_m1) or TBP (Hs00427620_m1).

## Western blot analysis

Protein samples were denatured for 5 min at 95 °C with Laemmli sample buffer, and protein concentration was measured by BCA Kit

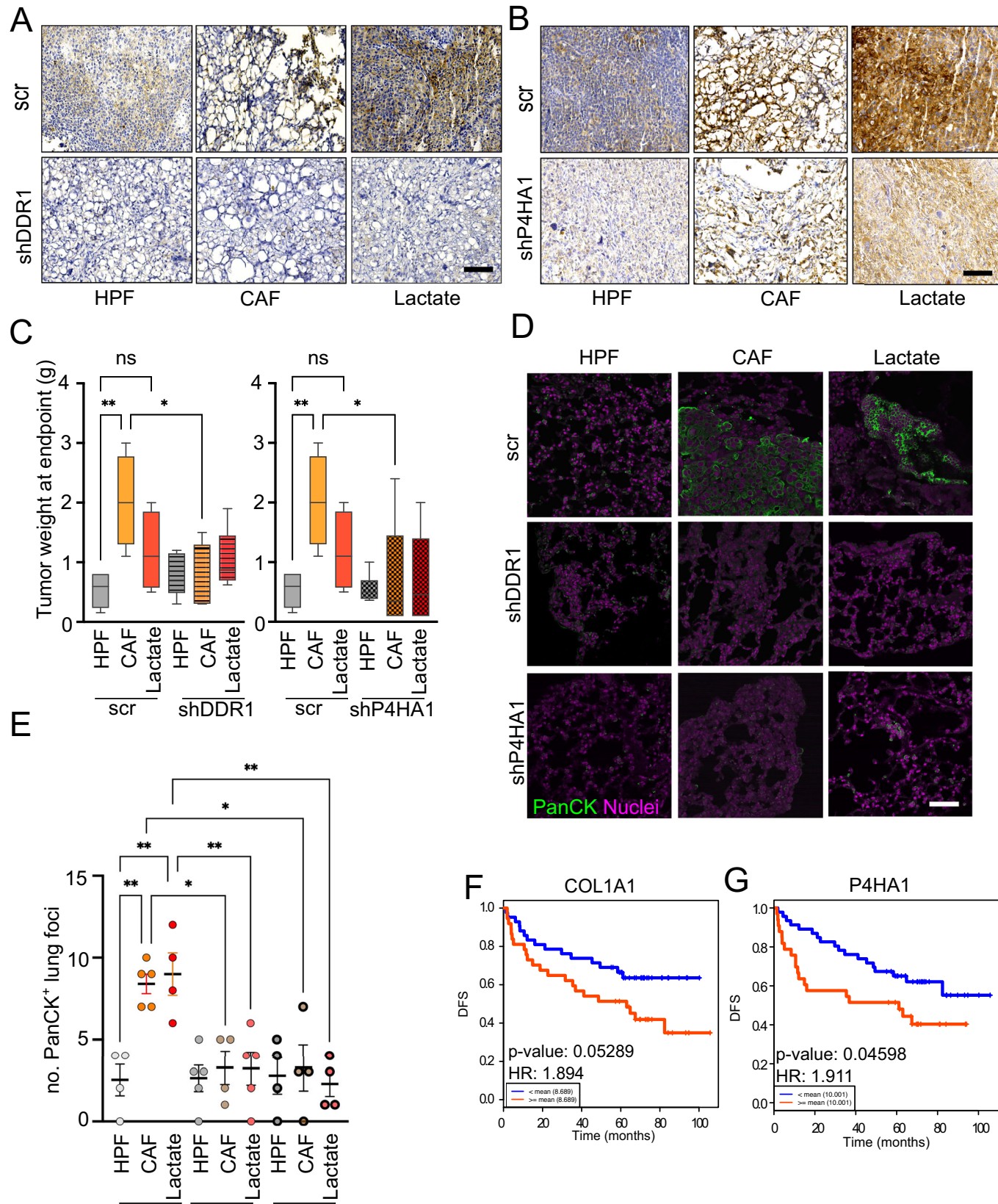

◄ **Figure 6. P4HA1-DDR1 axis sustains lactate-enhanced in vivo metastatic behavior of PCa cells.**

(A) Representative immunohistochemical detection of DDR1 in scrambled (scr) or shDDR1 xenografts tumors derived from the co-injection of DU145 cells with HPFs, CAFs or daily treated with lactate (1 g/kg). $n = 4$–5 mice/group. Scale bar: 100 μm, magnification ×20. (B) Representative immunohistochemical detection of P4HA1 in scrambled (scr) or shP4HA1 xenografts tumors. Scale bar: 100 μm, magnification ×20. (C) Means of tumor weights on day 60 from mice bearing scrambled or shP4HA1/shDDR1 tumors as indicated. Box plot: centerlines show the medians; box limits indicate the 25th and 75th percentiles; and whiskers extend to the minimum and maximum. (D) Representative immunofluorescence images for human Pancytokeratin (green) assessed in lung tissues derived from scrambled, shP4HA1 and shDDR1 xenografted mice. Nuclei were stained with DAPI (pseudocolor magenta). Scale bar: 10 μm. (E) Quantification of pancytokeratin-positive metastatic foci (>10 cells). (F, G) Kaplan–Meier disease-free survival analysis of PCa patients in the Glinsky cohort ($n = 79$) stratified by COL1A1 (F) or P4HA1 (G) mRNA abundance. HR: hazard ratio (95% confidence interval). Data information: bar graphs in (C, E) represent means ± SEM, $n = 4$–5 mice/group. Significance was determined using one-way ANOVA, followed by Tukey's multiple comparisons test (ns not significant; *$P < 0.05$; **$P < 0.01$). Source data are available online for this figure.

(Millipore, #71285-M). Samples (20–25 μg) were then loaded on 4–20% acrylamide precast SDS-PAGE gels (Bio-Rad, #456–8096) and then transferred on polyvinylidene difluoride membrane by Trans-Blot Turbo Transfer Pack (Bio-Rad, #1704157). The membranes were activated with methanol and then incubated in 5% milk or BSA with the appropriate antibodies overnight at 4 °C. The following primary antibodies used in this study: rabbit anti-P4HA1 (1:1000; HPA007599; Merck), p-DDR1 (1:500; #11994), DDR1 (1:1000; #5583), p-STAT3 (1:1000; #9145) (Cell Signalling Technology), COL1A1 (ab34710, 1:500 Abcam), STAT3 (1:500; sc-482; Santa Cruz Biotechnology,) and mouse anti-β actin and GAPDH (1:1000; sc-32233, sc-47778; Santa Cruz Biotechnology). Membranes were then incubated with the corresponding secondary antibody for 1 h at room temperature and visualized with Amersham AI600 Chemoluminescer Imager.

## Immunofluorescence staining

Prostate cancer cells were seeded onto glass coverslips ($1$–$2 \times 10^5$ per well of a six-well plate) and treated as indicated. Then cells were fixed with 4% formaldehyde or ice-cold methanol for 10 min (min). For nuclei staining, fixed cells were incubated with DAPI (Thermo Fisher Scientific #D3571) for 10 min at room temperature. Sample images were acquired using TCS SP8 microscope (Leica Microsystems) with LAS-AF image acquisition software. Samples were then permeabilized using 0.1% Triton X-100 for 15 min. PBS-based buffer containing 2% horse serum and 5% BSA was used as blocking agent for 1 h. Primary antibodies (1:100) were incubated overnight at 4 °C. After three washes of PBST (0.1% Tween 20), fluorescence-labeled secondary antibodies (1:1000, Thermo Fisher Scientific) were added and incubated for 1 h at room temperature. For tumor tissues immunofluorescence, the FFPE samples were subjected to Leica BOND-MAX™ automated system (Leica Microsystems)—washing steps and primary antibodies: pan-Cytokeratin (#ab234297, Abcam) and Col1a1 (#72026, Cell Signalling Technology), then Alexa Fluor-488 (#A-11008, Thermo Fisher Scientific) rabbit secondary antibody was incubated for 1 h at room temperature. Three to five images were captured using a TCS SP8 microscope (Leica Microsystems) with LAS-AF image acquisition software.

## ECM preparation

Cells were seeded and then treated as indicated, for 5–7 days to allow matrix deposition. The ECM was decellularized with an extraction buffer (20 mM $NH_4OH$, 0.5% Triton X-100 in PBS) until no intact cells were visible but the ECM remained on the dish. ECM was twice washed with PBS and then used for confocal imaging (DAPI was used to confirm no presence of cells), or for cell seeding.

Collagen deposition was estimated from fluorescent images and quantified by using Fiji software.

## Invasion assay

Invasion assay was performed using an 8-μm-pore transwell (#3428; Corning) coated with reconstituted Matrigel (#356334; BD Biosciences). Cells were seeded ($1 \times 10^5$/well), in the upper chamber of the transwell in serum-free medium and allowed to invade for 16 h toward complete medium. Air-dried membranes were stained with DiffQuick solution (#726443; BD Bioscience) and invasion was evaluated by counting invading cells to the lower surface of the filters (six randomly chosen fields).

## Transendothelial migration assays

HUVEC ($1 \times 10^5$ cells/well) were seeded on the 8-μm-pore transwell (#3428; Corning) in complete medium and cultured for 2–3 days to confluence. PCa cells ($5 \times 10^4$ cells/well) previously labeled with CellTracker Green (#C2925, Thermo Fisher Scientific) were added to the upper chamber in 200 μL of serum-free medium. An additional 500 μL of complete medium was added to each well of the lower chamber. Cells were incubated at 37 °C overnight and fixed by 100% methanol. Migrating PCa cells were quantified by TCS SP8 microscope (Leica Microsystems) with LAS-AF image acquisition software.

## Metabolomics

For selected ion monitoring (SIM) mode MS analysis, cells were scraped in 80% methanol, and phase separation was achieved by centrifugation at 4 °C. The methanol-water phase containing polar metabolites was separated and dried using a vacuum concentrator. Dried polar metabolites were dissolved in 20 μL of 2% methoxyamine hydrochloride in pyridine (Pierce-Thermo Fisher) and held at 37 °C for 2 h. After dissolution and reaction, 80 μL N-tert-Butyldimethylsilyl-N-methyltrifluoroacetamide with 1% tert-Butyldimethylchlorosilane MBTSTFA + 1% TBDMCS (Thermo Fisher) was added, and samples were incubated at 60 °C for 60 min. Gas chromatographic runs were performed with helium as carrier gas at 0.6 mL/min. The split inlet temperature was set to 250 °C and the injection volume of 1 μL. The GC oven temperature ramp was from 70 to 280 °C. The first temperature ramp was from 70 to 140 °C at 3 °C/min. The second temperature ramp was from 140 to 180 °C at 1 °C/min. Finally, the latest temperature ramp was from 180 to 280 °C at 3 °C/min. For the Quadrupole, an EI source (70 eV) was used. The ion source and transfer line temperatures were set, respectively, to 250 °C and 290 °C. For the determination

of relative metabolite abundances, the integrated signal of all ions for each metabolite fragment was normalized by the signal from norvaline, and cell number or protein content.

## Hydroxyproline assessment

Cells were washed once in PBS and hydrolyzed for 3.5 h at 130 °C in 6 N HCl. Thereafter, samples were vacuum-evaporated and dissolved in demineralized water for GC-MS analysis of hydroxyproline content per sample, which was finally normalized to the protein content of a parallel sample.

## Radioactive incorporation of lactate

Lactate uptake was evaluated by incubating the cells with uptake buffer solution (140 mmol/L NaCl, 20 mmol/L HEPES/Na, 2.5 mmol/L MgSO4, 1 mmol/L CaCl2, and 5 mmol/L KCl, pH 7.4) containing 14 C(U)-Lactic Acid (NEC599, PerkinElmer) for 15 min. Cells were subsequently washed with ice-cold PBS and lysed with 0.1 mol/L NaOH. All the radioactive signals were normalized on protein content.

## RNA sequencing analysis

The RNA sequencing (RNA-seq) analysis was performed to investigate the expression profiles of DU145 cells as in (Ippolito et al, 2022a, 2022b). For each condition, four technical replicates were produced. Total RNA was extracted using RNeasy Kit (#74104, Qiagen) according to the manufacturer's instructions. RNA quantity and quality were evaluated with a NanoDrop 1000 Spectrophotometer (Thermo Fisher Scientific). RNA-seq for all experiments was performed at the Institute of Oncology Research (Bellinzona, Switzerland) using Next Ultra II Directional RNA Library Prep Kit for Illumina starting from 800 ng of total RNA each sample and sequenced on the Illumina NextSeq500 with single-end, 75 bp long reads.

## Prostaspheres formation assay

In total, $1 \times 10^3$ cells/mL were seeded in a six-well plate coated with 12 mg/mL polyhydroxyethylmetalcrylate (PolyHEMA). Overall, $1 \times 10^3$ cells were resuspended in 1 ml of DMEM/F-12 (1:1) (#11039021 GibcoTM) supplemented with 0.1 mg/mL FGF, 10 µg/mL EGF, 37 ug/mL insulin, 100X CTS N-2 Supplement (#A1370701 GibcoTM), 50× B27 Supplement (#17504044 GibcoTM), 2 mmol/L L-glutamine and 1% penicillin/streptomycin, allowing prostaspheres formation. Cells were maintained at 37 °C and 5% $CO_2$ for 10 days. Number and size of prostaspheres with ≥80 µm diameter were then assessed and representative pictures captured.

## In vivo experiments

Six to 8-week-old male SCID mice (NOD.CB17-Prkdcscid/ NCrHsd, Envigo) were housed and used at the Centro Stabulazione Animali da Laboratorio (CESAL, Florence, Italy) under sterile conditions with ad libitum access to food and water. For lung retention assay, $0.5 \times 10^6$ pre-treated siCTR and $0.5 \times 10^6$ siP4HA1 PCa cells were labeled with CellTracker Green (C7025; Thermo Fischer Scientific) and Red (C34552; Thermo Fischer Scientific), respectively, 1:1 mixed and

co-injected into the lateral tail vein of four randomized mice per group. Mice were euthanized after five hours and lungs were imaged with TCS SP8 microscope (Leica Microsystems) with LAS-AF image acquisition software. For spontaneous metastasis assay, $1 \times 10^6$ cells were resuspended in 100 µl PBS and subcutaneously co-injected with $0.5 \times 10^6$ HPFs or CAFs into the flanks of SCID mice. In total, $1 \times 10^6$ cells without fibroblasts were injected for the lactate group (i.p., 1 g/kg/ daily). Animal work was carried out under the Project license n. 591/ 2022-PR, and was approved by the Italian Health Minister. All the procedures were conducted in accordance with the protocols approved by the Institutional Animal Care and Use Committee of the University of Florence.

## Immunohistochemistry (IHC)

Individual specimens from mouse tumors/lungs were dissected and fixed in 4% PFA for histology and IHC analysis. For prostate tissue analyses, we used a tissue microarray (#PR633a) purchased from US Biomax. Sections were stained with P4HA1 (1:100, HPA007599; Merck)) and DDR1 (1:100, #5583; Cell Signaling Technology). Immunohistochemistry was performed using the Leica BOND-MAX™ automated system (Leica Microsystems). Slides were developed with 3′3-diaminobenzidine and counterstained with hematoxylin. Staining intensity for P4HA1 or DDR1 was analyzed independently by at least two researchers and given a score of 0, 1 +, 2 +, 3 +. In all, 5-µm lungs sections were cut and stained using hematoxylin and eosin (HE) for pathological assessment. The specificity of the antibodies was established using negative and positive human tissue samples. For antibodies used on mouse tissues, the blocking Mouse on Mouse (M.O.M.) basic kit (Vector Laboratories, BMK-2202) was used according to the manufacturer's datasheet. Images were acquired by using a slide scanner (Aperio LV1; Leica Biosystems) and analyzed with software ImageScope (RRID:SCR_020993).

## Statistical analysis

Correlation analysis was performed using patient raw data from the web-based interface Cancertool (www.cancertools.org) or GEPIA (http://gepia.cancer-pku.cn). Statistical analysis was performed using GraphPad Prism v9 (GraphPad Software Inc., CA, USA) on $n \geq 3$ biological replicates. Error bars represent the standard error of the means (SEM). Comparison between the two groups was performed using an unpaired two-tailed Student's $t$ test. One-way ANOVA followed by a post hoc Tuckey test was performed to compare more than two groups. $P$ values < 0.05 were considered significant. $P < 0.05$ was considered statistically significant. $P < 0.05$ is denoted with *, ≤0.01 with **, ≤0.001 with ***, and ≤0.0001 with ****; $P \geq 0.05$ is considered not significant ('ns').

# Data availability

This study includes no data deposited in external repositories.

The source data of this paper are collected in the following database record: biostudies:S-SCDT-10_1038-S44319-024-00180-z.

# Peer review information

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

## Acknowledgements

The work was funded by Associazione Italiana Ricerca sul Cancro (AIRC) and Fondazione Cassa di Risparmio di Firenze (grant Multiuser 19515 to PC and AM) and AIRC (grant IG 22941 to AM, grant IG 24731 to PC), Swiss Cancer League (KLS-4899-08-2019 to CVC) and Fondazione Ticinese Ricerca sul Cancro. LI is supported by a Fondazione Pezcoller Foundation fellowship. NL and EP (project code 26599) are supported by AIRC fellowships. The data presented in the current study were in part generated using grants by European Union, National Recovery and Resilience Plan, Mission 4 Component 2— Investment 1.4—National Center for Gene Therapy and Drugs based on RNA Technology—NextGenerationEU—Project Code CN00000041-CUP B13C22001010001 (to EG) and Creation and strengthening of "innovation ecosystems", construction of "territorial R&D leaders"—TUSCANY HEALTH ECOSYSTEM (THE) NextGenerationEU—Project Code ECS_00000017-CUP B83C22003920001 (to PC). We thank Dr. Matteo Parri for technical support at the metabolomics facility. The illustration was created with BioRender.com.

## Author contributions

**Luigi Ippolito**: Conceptualization; Investigation; Writing—original draft; Writing—review and editing. **Assia Duatti**: Formal analysis; Visualization; Methodology. **Marta Iozzo**: Data curation; Investigation; Methodology. **Giuseppina Comito**: Investigation; Methodology. **Elisa Pardella**: Data curation; Investigation; Methodology; Writing—review and editing. **Nicla Lorito**: Investigation. **Marina Bacci**: Investigation. **Erica Pranzini**: Investigation; Methodology; Writing—review and editing. **Alice Santi**: Investigation. **Giada Sandrini**: Software. **Carlo V Catapano**: Visualization; Methodology. **Sergio Serni**: Resources. **Pietro Spatafora**: Resources. **Andrea Morandi**: Conceptualization; Methodology; Writing—review and editing. **Elisa Giannoni**: Supervision; Writing—original draft; Project administration; Writing—review and editing. **Paola Chiarugi**: Supervision; Funding acquisition; Writing—original draft; Project administration; Writing—review and editing.

Source data underlying figure panels in this paper may have individual authorship assigned. Where available, figure panel/source data authorship is listed in the following database record: biostudies:S-SCDT-10_1038-S44319-024-00180-z.

## Disclosure and competing interests statement

The authors declare no competing interests.

# Expanded View Figures

**Figure EV1. Collagen signature is enhanced in CAF-derived LA-treated PCa cells.**

(A) Dotplot showing Reactome-enrichment of genes associated to the cell ECM pathway in LA-treated DU145 cells. Circle size is proportional to the enrichment score. Red dots highlight the significantly enriched pathways. (B) qRT-PCR analysis for COL1A1 mRNA level in DU145 and PC3 cells treated with HPF-CM, CAF-CM or LA. (C) Immunofluorescence analysis of extracellular COL1A1 on decellularized matrix derived from DU145 treated as indicated ± MCT1i; scale bar: 10 μm. Quantification of fluorescence signal was reported. (D) Immunofluorescence analysis of intracellular COL1A1 on DU145 cells, silenced for MCT1 (siMCT1) or a non-targeting control (siCTR). Nuclei (blue) were stained with DAPI. Scale bar: 10 μm. Box plot showing the quantification of fluorescence signal per cell: centerlines show the medians; box limits indicate the 25th and 75th percentiles; and whiskers extend to the minimum and maximum. $n = 9\text{-}27$ cells from three biological replicates. (E) Enrichment plots of the Hallmark Hypoxia showing a positive association between this MSigDb datasets and the LA-exposed DU145 gene expression profile. NES, normalized enrichment score. (F) Representative western blot analysis of P4HA1 in 22Rv1 cells, exposed to LA and to a serum-free medium as control (not treated, NT). GAPDH was used as loading control. Representative western blot analysis of P4HA1 in PC3 cells treated as indicated ± MCT1i. Beta-actin was used as loading control. (G) Representative western blot analysis of P4HA1 and MCT1 in siCTR and siMCT1-DU145 cells. Beta-actin was used as loading control. Data information: bar graphs in (B–D) represent means ± SEM of $n = 3$ biological replicates. Significance was determined using one-way ANOVA, followed by Tukey's multiple comparisons test (*$P < 0.05$; **$P < 0.01$; ***$P < 0.001$; ****$P < 0.0001$). Source data are available online for this figure.

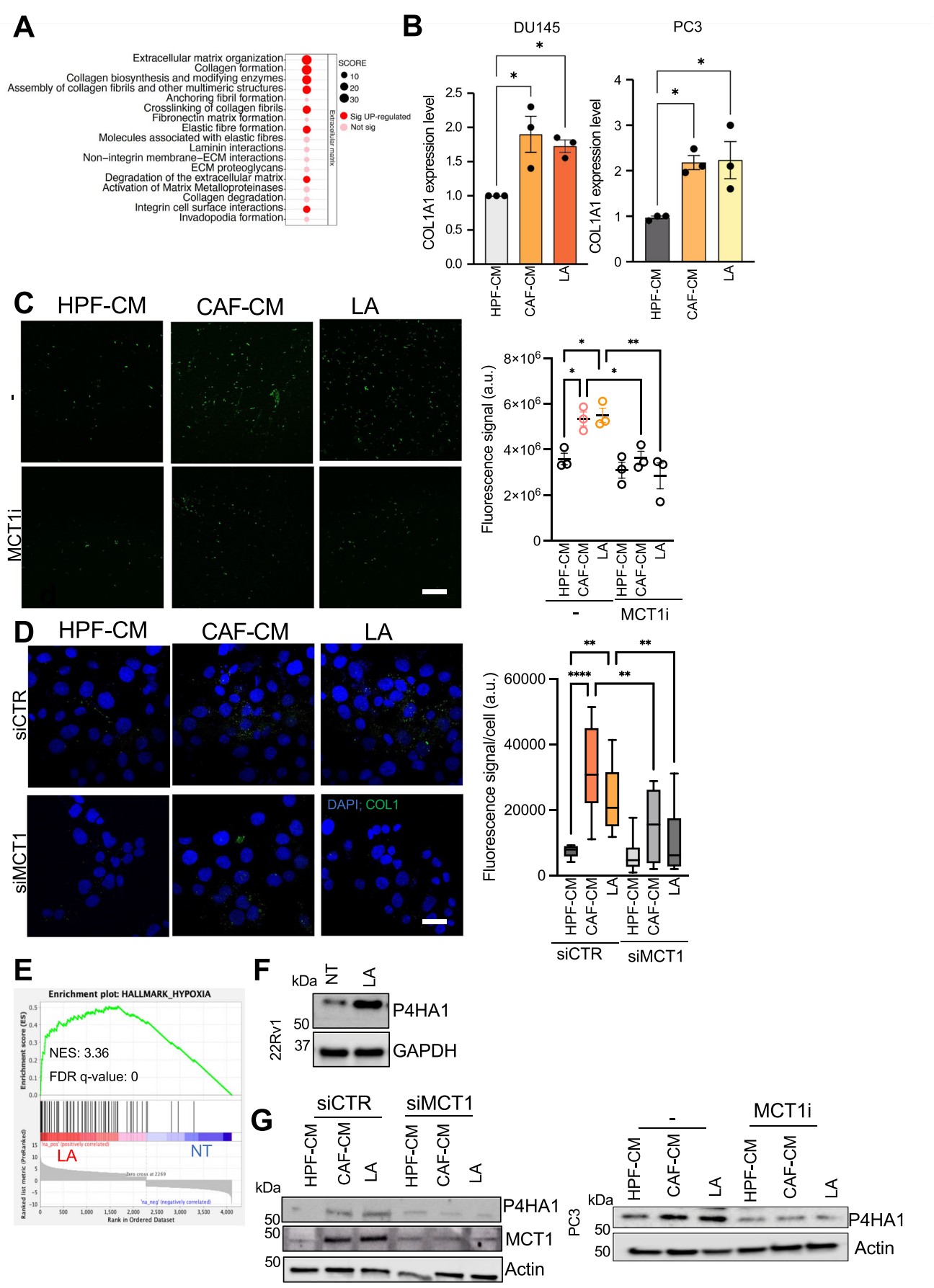

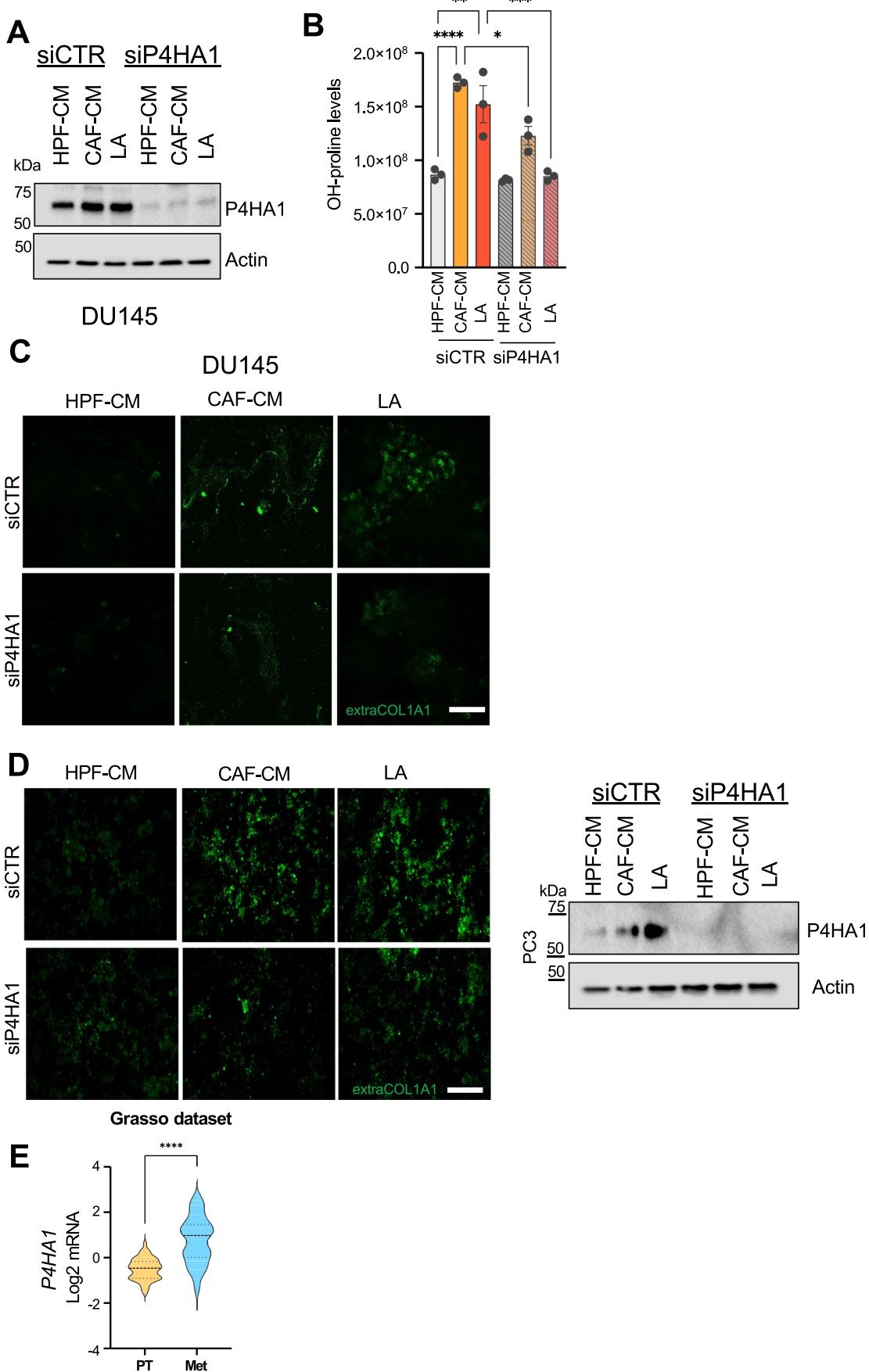

◀ **Figure EV2. P4HA1 sustains the LA-induced collagen deposition in PCa cells.**

(A) Representative western blot analysis of P4HA1 in DU145 silenced for P4HA1, treated as indicated. Beta-actin was used as loading control. (B) GC-MS analysis of hydroxyproline levels in siCTR and siP4HA1-DU145 cells, treated as indicated. (C, D) Immunofluorescence analysis of extracellular COL1A1 on decellularized matrix derived from DU145 (C) and PC3 (D) cells, treated as indicated and silenced for P4HA1; scale bar: 10 μm. (E) Violin plots representing P4HA1 mRNA expression in primary (PT) and metastatic (Met) tumor specimens of Grasso dataset ($n = 88$). Data information: bar graphs in (B, E) represent means ± SEM; (B) $n = 3$ biological replicates. Significance was determined using one-way ANOVA, followed by Tukey's multiple comparisons test (B) (*$P < 0.05$; **$P < 0.01$; ***$P < 0.001$; ****$P < 0.0001$), or unpaired two-tailed $t$ test (E) (****$P < 0.0001$). Source data are available online for this figure.

**A**

No. PC3 invading cells/fov

HPF-CM / CAF-CM / LA — siCTR

HPF-CM / CAF-CM / LA — siP4HA1

\* *** **** **

**B**

No. PC3 invading cells/fov

HPF-CM / CAF-CM / LA — -

HPF-CM / CAF-CM / LA — DHB

\* ** *

**C**

No. DU145 invading cells/fov

HPF-CM / CAF-CM / LA — -

HPF-CM / CAF-CM / LA — DHB

** ** * *

**D**

no. 22Rv1 invading cells/fov

NT / LA — -

NT / LA — DHB

*** ***

**E**

no. prostaspheres/1000 cells

NT / LA — -

NT / LA — DHB

** **

NT    LA

-

DHB

22Rv1

**F**

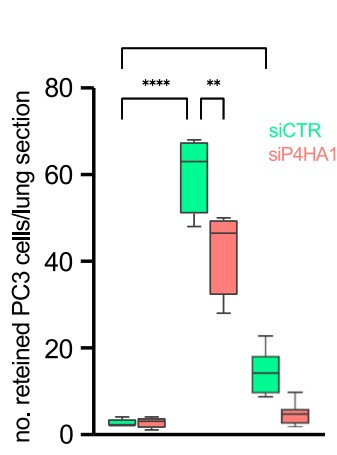

no. reteined PC3 cells/lung section

siCTR

siP4HA1

HPF-CM    CAF-CM    LA

**** **

**Figure EV3.   Targeting LA-induced P4HA1 impairs PCa cell invasive and colonizing abilities.**

(A) Invasion assay performed on PC3 cells treated as indicated, silenced for P4HA1. (B) Invasion assay performed on PC3 cells treated as indicated, ± DHB (20 μM). (C) Invasion assay performed on DU145 cells treated as indicated, ± DHB (20 μM). (D) Invasion assay performed on 22Rv1 cells treated as indicated, ± DHB (20 μM). (E) Prostaspheres formation assay (representative pictures at day 10 are shown, scale bar: 100 μm) for 22Rv1 cells treated as indicated, ± DHB. (F) Lung retention assay performed on PC3 cells, treated as indicated, then labeled with CellTracker dyes (green for siCTR and red for siP4HA1) before being 1:1 injected into the tail vein of SCID mice. Quantification of green and red fluorescent cells per lung section field of view was reported as box plot: centerlines show the medians; box limits indicate the 25th and 75th percentiles; and whiskers extend to the minimum and maximum; $n = 4$–6 sections. Data information: bar graphs in (A–F) represent means ± SEM, (A–E) $n = 3$ biological replicates, (F) $n = 4$ mice/group. Significance was determined using one-way ANOVA, followed by Tukey's multiple comparisons test (*$P < 0.05$; **$P < 0.01$; ***$P < 0.001$; ****$P < 0.0001$). Source data are available online for this figure.

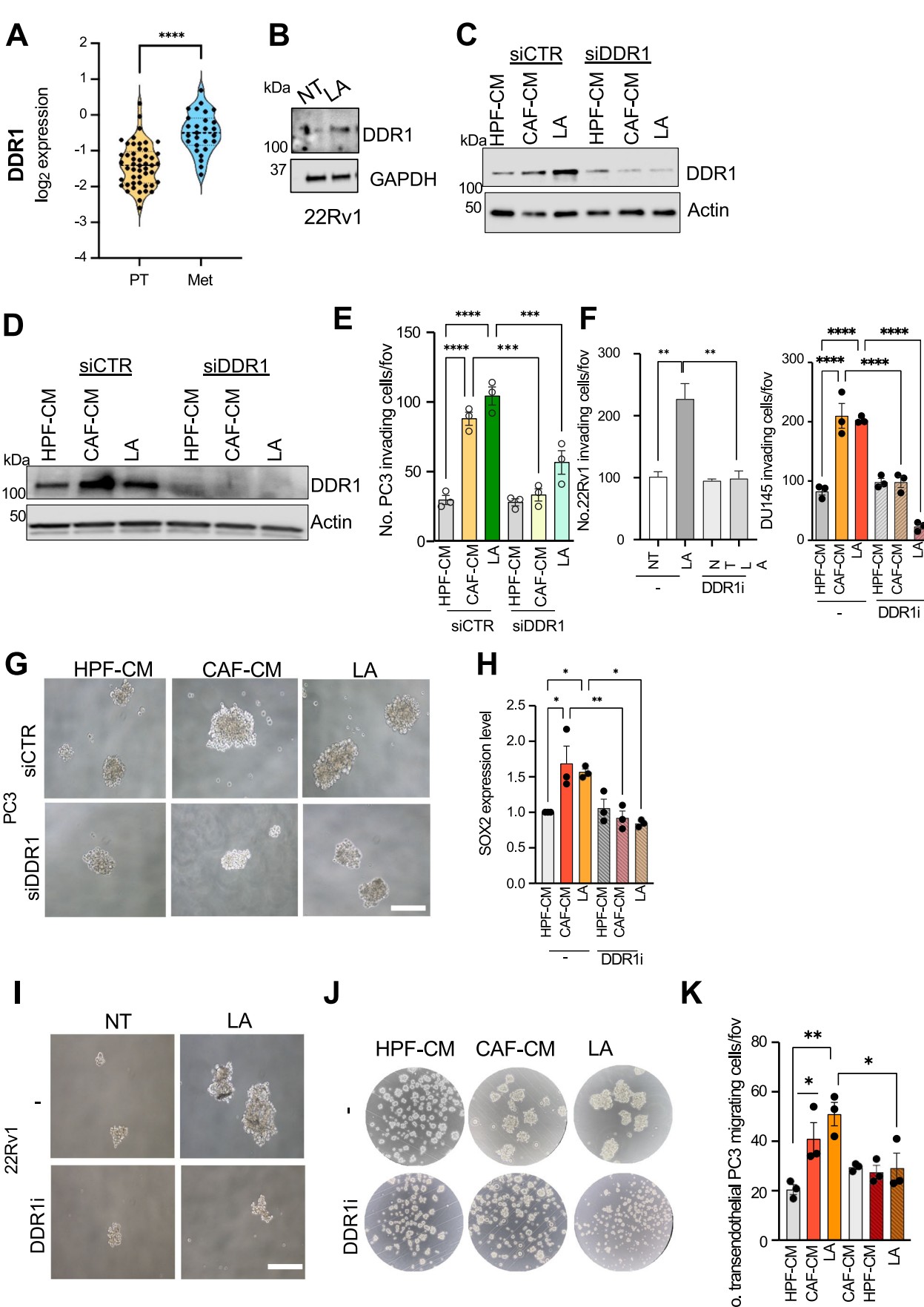

◄ **Figure EV4. DDR1 activation is linked to the LA-sustained aggressiveness in PCa cells.**

(A) Violin plots representing DDR1 mRNA expression in primary (PT) and metastatic (Met) tumor specimens of Grasso dataset ($n = 88$). Each plot means an individual specimen. (B) Representative western blot analysis of DDR1 in 22Rv1 cells treated or not (NT) with LA. GAPDH was used as loading control. (C, D) Representative western blot analysis of DDR1 in siCTR and siDDR1 DU145 and PC3 cells treated as indicated. Beta-actin was used as loading control. (E) Invasion assay performed on PC3 cells silenced for DDR1, treated as indicated. (F) Invasion assay performed on 22Rv1 and DU145 cells treated as indicated, ± 7rh (DDR1i, 250 and 500 nM, respectively). (G) Representative pictures for prostaspheres formation in PC3 cells silenced for DDR1, treated as indicated. Scale bar: 100 μm. (H) qRT-PCR analysis for SOX2 mRNA level in DU145 cells treated as indicated, ± DDR1i (500 nM). (I–L) Representative pictures for prostaspheres formation in 22Rv1 (I) and DU145 cells (J), treated as indicated, ± DDR1i. Scale bar: 100 μm. (K) Transendothelial migration performed on DU145 cells treated as indicated, ± DDR1i. The number of cells migrating were quantified per field of view (FOV). Data information: bar graphs in (A, E–K) represent means ± SEM; (E–K) $n = 3$ biological replicates. Significance was determined using unpaired two-tailed $t$ test (A) (****$P < 0.0001$), or one-way ANOVA, followed by Tukey's multiple comparisons test (E–K); (*$P < 0.05$; **$P < 0.01$; ***$P < 0.001$; ****$P < 0.0001$). Source data are available online for this figure.

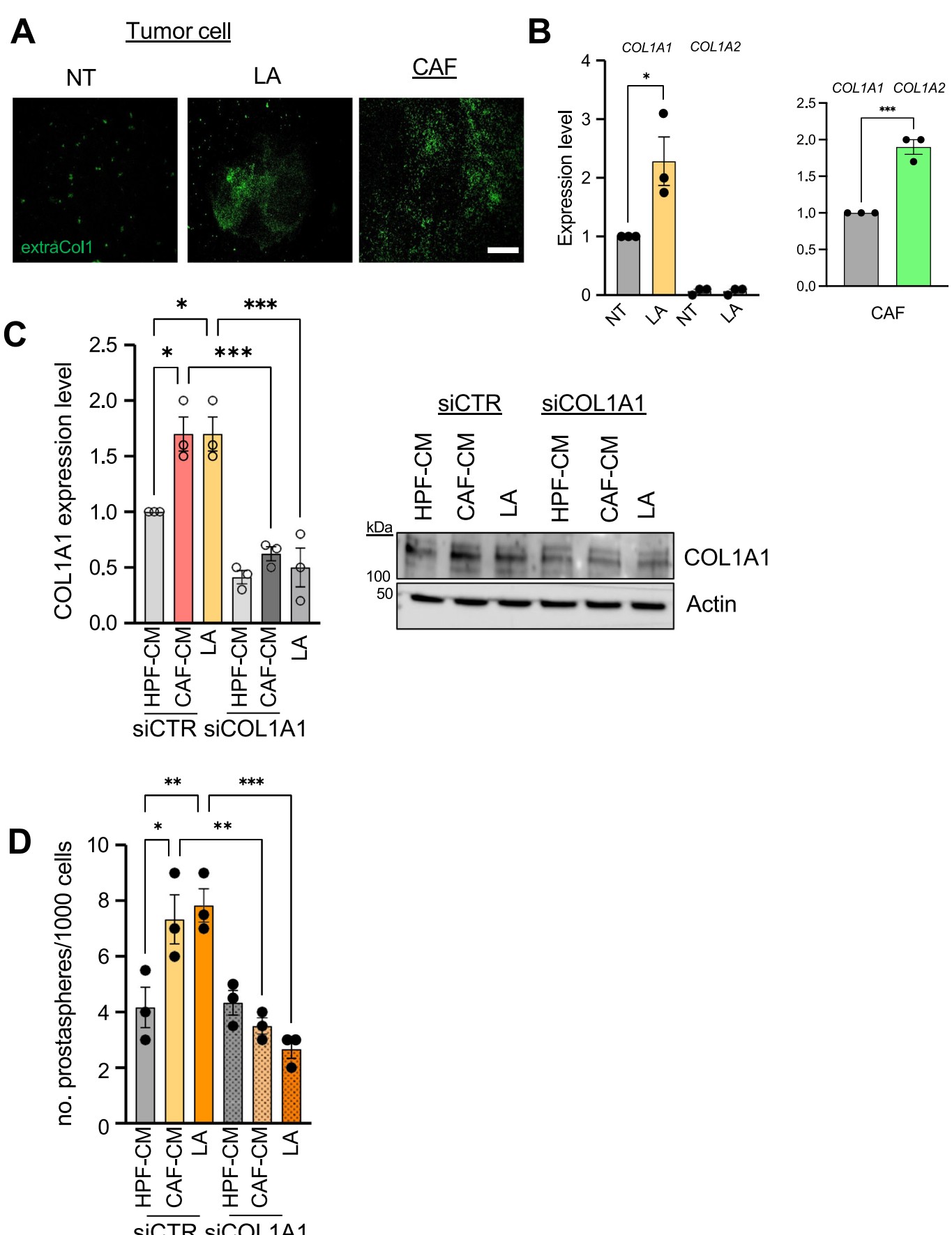

**Figure EV5.  Tumor-derived collagen boosts LA-induced aggressive traits in PCa cells.**

(A) Immunofluorescence of extracellular collagen I on decellularized matrices derived from tumor cells, treated with LA for 4 days or not (NT), and from CAFs. (B) qRT-PCR analysis for COL1A1 and COL1A2 in PC3 cells treated or not with LA, and in three different CAFs (each point represents a CAF specimen). (C) qRT-PCR analysis for COL1A1 and western blot analysis of COL1A1 in PC3 cells, silenced for COL1A1 and treated as indicated. (D) The number of the prostaspheres-derived PC3 cells silenced for COL1A1, treated as indicated, was quantified and plotted. Scale bar: 100 μm. Results are shown as means ± SEM. Data information: bar graphs in (B–D) represent means ± SEM, $n = 3$ biological replicates. Significance was determined using one-way ANOVA, followed by Tukey's multiple comparisons test (B–D); *$P < 0.05$, **$P < 0.01$, ***$P < 0.001$. Source data are available online for this figure.

