## [Peer Review File · EMBO Reports]

Lactate supports cell-autonomous ECM production to sustain metastatic behavior in prostate cancer

Luigi Ippolito, Assia Duatti, Marta Iozzo, Giuseppina Comito, Elisa Pardella, Nicla Lorito, Marina Bacci, Erica Pranzini, Alice Santi, Giada Sandrini, Carlo Catapano, Sergio Serni, Pietro Spatafora, Andrea Morandi, Elisa Giannoni, and Paola Chiarugi

Corresponding author(s): Paola Chiarugi (paola.chiarugi@unifi.it), Luigi Ippolito (luigi.ippolito@unifi.it)

Review Timeline:

Submission Date:	12th Jul 23
Editorial Decision:	8th Sep 23
Appeal Received:	12th Sep 23
Editorial Decision:	15th Sep 23
Revision Received:	22nd Dec 23
Additional Correspondence:	14th Feb 24
Editorial Decision:	23rd Feb 24
Revision Received:	20th Apr 24
Editorial Decision:	24th May 24
Revision Received:	30th May 24
Accepted:	7th Jun 24

Editor: Deniz Senyilmaz Tiebe

Transaction Report:

Dear Prof. Chiarugi,

Thank you for submitting your manuscript to EMBO Reports. We have now received three referee reports, which are included below.

I apologize for this unusual delay in getting back to you. It took longer than anticipated to receive the full set of referee reports given this busy time of the year.

We concur with the referees that the proposed role of CAF derived lactate in promoting collagen deposition and metastatic behavior of prostate tumors is in principle very interesting. However, they also raise concerns that preclude publication in this journal. They point out concerns regarding the methodologies used and conclusiveness of the dataset. As such, they do not recommend publication in EMBO Reports. Given such input from these recognized experts who are also experienced referees, and the amount of experimentation required to address these concerns, we cannot offer further consideration of your manuscript.

However, in case you feel that you can address the referee concerns in a timely and thorough manner, and can obtain data that would considerably strengthen the study as in the referee reports, we would have no objection to consider a revised manuscript (along with a point-by-point response to the referee concerns) in the future. Please note that if you were to send a new manuscript this would be assessed again with respect to the literature and the novelty of your findings at the time of resubmission and in case of a positive editorial evaluation, the manuscript would be sent back to the original referees. I would like to emphasize that we will be reluctant to approach the referees again in the absence of major revisions, and we need strong support from the referees to consider publication here.

Thank you in any case for the opportunity to consider this manuscript. I am sorry that I cannot communicate more positive news, but nevertheless hope that you will find our referees' comments helpful.

Kind regards,

Deniz Senyilmaz Tiebe

Deniz Senyilmaz Tiebe, PhD
Editor
EMBO Reports

Referee #1:

In this manuscript, Ippolito et al. study tumor-stroma interactions in prostate cancer (PCa) and propose that lactate secreted by cancer-associated fibroblasts (CAFs) promotes stem cell properties and invasiveness of PCa cells via a crosstalk involving collagen production and P4HA1-DDR1-STAT3 pathway. The authors show that CAF-conditioned medium and lactate activate secretion of collagen by PCa cells which leads to the acquisition of a pro-metastatic program. The idea that CAF-derived lactate promotes secretion of collagen implicated in PCa metastatic phenotype is novel and would be of interest to the cancer research community. However, the study suffers from a number of important issues and in places lacks rigor. Details below.

Major Points:

1. The investigators propose CAFs as a source of secreted lactate in PCa leading to an interesting mechanistic insight into collagen production and modification to promote stem cell properties and invasion in PCa cells. Importantly, cancer cells can be a major source of lactate due to excessive glucose consumption and glycolysis (Warburg effect). Yet, this is not even mentioned by the authors. How significant are CAFs as a source of secreted lactate in tumors? Lactate levels from CAFs could simply be "a drop in the ocean" compared to other sources, such as cancer cells. This should be addressed. Adding conditioned medium from cultured CAFs to PCa cells does not establish CAFs as a significant lactate source within PCa tumors.
2. The investigators show that lactate can promote collagen production in PCa cells. However, this is only shown in vitro. This should also be addressed in vivo. Is col1a1 deposition increased in the mouse tumor models where CAFs are co-implanted, or mice are treated with lactate?
3. Figure 1e shows that CAF-CM and lactate can promote P4HA1 expression in PCa cells. However, whereas MCT1i leads to substantial P4HA1 repression in lactate samples, the same cannot be said about CAF-CM samples, which show a modest repression at best. This is essentially ignored by the authors but is important since the study proposes that stromal lactate enhances collagen synthesis by increasing P4HA1 expression via MCT1 transporter. Does this discordance undermine the idea that lactate from the CAF-CM is promoting P4HA1 in PCa cells? Could CAF-CM induction of P4HA1 be lactate independent?

4. Based on Figure 2d and Supplementary Figure S3f, the authors state that "P4HA1 is required for an efficient survival in the circulation, extravasation, and retention in the secondary sites". However, it is not clear what mouse experiments in Figure 2d and Supplementary Figure S3f are really measuring. How was survival in circulation measured? What about extravasation? The investigators analyze the cell numbers in lungs 5 hours after intravenous injection. Is this enough time for extravasation? Were the cancer cells that were quantified in lungs, confirmed to be outside the vasculature in the lung parenchyma? There is also a major discordance between "representative" images and the quantifications. These experiments lack rigor.

5. In supplementary Figure 3a-b, there seems to be substantial experimental variations and lack of consistency. If experiments are repeated 3 times why would there be a 3-5 fold difference in invasion between control samples in 3A and 3B? This is surprising.

6. In several figures, representative images of are not really representative when compared to quantifications. For example, in Figure 2b, images shown on the left do not represent the quantification on the right. The number of cells in CAF-CM and lactate condition with siP4HA1 cells appears to be much higher than the HPC-CM condition with siP4HA1 cells (not the case in the quantification). Also, in Figure 4d, images on the left show sphere sizes rather than sphere number which is quantified on the right.

7. Supplementary Figure 6c-d clearly shows that CAF-CM still phosphorylates pSTAT3 in P4HA1 knockdowns and cells treated with P4HA1 inhibitor, contrary to what the authors describe (lines 232-234). Please explain.

8. The experiment presented in Figure 4b is very confusing. What concrete conclusions can be drawn from this? This should be explained thoroughly or removed.

9. In Figure 6D, it is unclear what is really metastasis in the lung images. Immunostaining of a PCa cell marker would provide the specificity required.

10. Quantification is missing in several immunofluorescence analyses (Figure 1c, S1c-d, 5f, S7e).

11. The quality of western blots is poor in several panels, such as Figure 3b, Figure 5c and 5g.

Minor Points:

1. For the table presented at the end of the manuscript, a table legend and further explanations in the text are needed.

2. In Figure 1c, it is not clear how long the PCa cells were exposed to HPF- CAF-CM or lactate. This is neither mentioned in the text nor the Methods. The same applies for all results obtained with conditioned medium or lactate supplementation.

3. In Figure 3a, a trendline is missing.

4. A model to visualize all the mechanistic players involved in the proposed findings, would be helpful to crystalize the take home message of the paper.

5. Figure labeling mix-ups.

line 193 : Figure 3g (not 3f)

line 196 : Figure 3h (not 3g)

Figure 1 legend : panels g and h are inversed. Moreover, panel h is wrongly termed f in the legend).

Supp Fig 1 legend : panels c, d and e completely mixed.

Supp Fig 3f legend : n=3 mice per group mentioned whereas n=4 in the Methods.

Supp Fig 4 and legend : panels not in alphabetical order.

Supp Fig7 : Last panel is f not g.

Referee #2:

Ippolito and colleagues propose that CAF-secreted lactate promotes the expression of prolyl-4-hydroxylase (P4HA1) in PCa cells, which in turn leads to de novo collagen-induced DDR1 activation and increased invasiveness. The work is elegantly carried out but lacks experimental data to make the context of the study more biologically relevant.

1) The paradigm of lactate-generating CAF developed in the current manuscript is different from data published by others where high lactate concentrations in tumors are used by CAF to support progression instead of CAF generating lactate which in turn

influences the phenotype of cancer cells.

See for instance: <https://www.ncbi.nlm.nih.gov/pmc/articles/PMC8606002/>

The CAF contribution to tumor lactate vs. the production of lactate by cancer cells is all the more unclear as hypoxic prostate cancer cells are largely glycolytic and locally produce high lactate concentrations in tumors. The authors themselves refer to the control of P4HA1 by the hypoxia-inducible transcription factor (HIF-1).

2) The use of CM may look straightforward but introduces bias. First, the extent of lactate concentration in CM from CAF and HPF is not indicated. Second, if differences result from a higher glycolytic turnover in CAF (than in HPF), there are several other metabolic intermediates that are different between the two CM. To which extent reduced glucose and increased (extracellular) pyruvate concentrations influence the phenotype of treated cells is unclear. Same comment for the buffering capacity of the medium which may be altered in CAF medium due to H⁺ release together with lactate.

Also, does addition of the CAF-CM amount of lactate in HPF-CM recapitulate the reported effects? Does the acidification of HPC-CM and reduced glucose concentration in HPC-CM fail to mimic these effects?

3) The use of lactic acid (as indicated in Methods) instead of lactate (indicated in Results) may account for some differences related to acidification. This should be addressed. Same for the concentrations used. Many papers in the field relate to a concentration of 20-40mM because of a wrong interpretation of old papers expressing the concentrations in mg/ml. The rationale for the lactate concentration should be supported by experimental data validating the chosen concentration for in vitro studies.

Specifics:

- Lack of changes in cell proliferation and in vitro survival upon P4H1, DDR1, COL1A1 and STAT3 silencing should be documented.

- Stimulatory effect of lactate on total STAT3 is visible in different panels of fig 5. The balance between changes in phosphorylated form and total abundance should be clarified.

- Injection (i.p) of lactate in mice may lead to systemic effects making interpretation of data confusing. The impact of serum lactate on tumor lactate concentration should be proved.

Minor:

- Fig 1C : contrast should be optimized to better see fluorescent signals

Referee #3:

Ippolito et al. provide novel metabolic insights in the metastatic behavior of prostate cancer. The authors uncovered a new mechanism in prostate cancer (PCa) cells, by which cancer-associated fibroblasts (CAF)-secreted lactate promotes expression of genes coding for the collagen family. In detail, lactate-utilizing PCa cells rely on α -ketoglutarate (α -KG), which activates the α -KG-dependent collagen prolyl-4-hydroxylase (P4HA1) to support collagen hydroxylation. In turn, de novo synthesized collagen exerts a signaling role by activating discoidin domain receptor 1 (DDR1), supporting stem-like and invasive features of PCa cells. The authors have shown nicely that the inhibition of lactate-induced collagen hydroxylation and DDR1 activation reduces metastatic colonization in vitro and in vivo. Interestingly, STAT3 activation is crucial to control the invasive and stem-like features of lactate-rewired PCa cells. Furthermore, the authors provide evidence that the lactate-dependent P4HA1/Col1/DDR1 axis is critical for STAT3 activation.

Overall, this is an important manuscript providing a new understanding of the P4HA1/Col1/DDR1 signaling axis regulating prostate cancer metastasis and link collagen remodeling/signaling and the nutrient environment exploited by PCa cells. The authors present their findings clearly and describe them accurately within the text. I really enjoyed reading this paper. A real strength of this study is the identification of a unique target to block metastasis and that the paper connected the metabolic and molecular consequences and presented novel therapeutic approach.

Below are comments that aim to further increase the clarity and significance of this study:

1. Does the expression of MCT1 increase when Du145 cells are exposed to CAF-CM or lactate? Can the authors evaluate the efficiency of MCT1 inhibition in tumor cells?
2. Can the authors comment whether the pH of the medium was adjusted when supplemented with 20 mmol/L lactate.
3. Do P4HA1-silenced PCa cells metastasize to other organs other than the lung (Fig. 2D)
4. Can the authors speculate/discuss how lactate conditioning promotes DDR1 expression?
5. The authors displayed nicely that STAT3 inhibition negatively impacts CAF/lactate enhanced invasiveness and prostaspheres formation. Can the authors provide evidence that STAT3 overexpression impacts CAF/lactate invasiveness? Does STAT3 overexpression lead to a reversed effect on invasiveness and prostaspheres formation?

6. The authors displayed representative western blot images, indicating that lactate/driven DDR1 and Col1 expression are significantly reduced in STAT3-silenced or STAT3-inactive PCa cells exposed to stromal lactate. However, representative western blot images do not give statistical information about changes in the protein abundance. In order to provide evidence on significant changes in the protein abundance, could the authors add a statistical western blot analysis to the respective figure or in the supplemental part?

7. Could the authors give more information about the RNA sequencing method? How many reads per samples were measured? What reference genome was used in order for subsequent alignment and which software was used? Did the authors correct the resulting p-values for multiple testing?

8. It would be helpful to add a graphical abstract or a final figure in order to summarize the novel identified P4HA1/Col1/DDR1 signaling axis that regulates prostate cancer metastasis

Minor comments:

Line 174: Reference is stated as PMID and should be edited

Line 193 (Figure 3f): Figure 3f displays ALDH1A1 expression, not the ability to intra/extravasate sustained by lactate

Line 195 (Figure 3g): Figure 3g shows fluorescently labeled DH145 cells, not DDR1 expression in Gleason-graded PCa patient samples. Figure 3h is not described in the results section.

Line 259: There is no supplemental Figure S7f

Line 401: Could the authors add the sequences of all used siRNAs?

** As a service to authors, EMBO Press provides authors with the ability to transfer a manuscript that one journal cannot offer to publish to another journal, without the author having to upload the manuscript data again. To transfer your manuscript to another EMBO Press journal using this service, please click on Link Not Available

Dear Editor,

Thanks for your kind reply concerning the referees' comments of our submitted manuscript.

We highly appreciated that our study on lactate & collagen in prostate tumor aggressiveness has been read with enthusiasm and interest. Of course, we think the comments from referees are strongly helpful for us, in order to identify the points needing to be more accurate and substantial from a methodological point of view, to further ameliorate the scientific robustness of our manuscript, thus positively affecting the overall message.

Alongside thanking the referees and agreeing with most of their suggestions, we will be pleased to address their concerns. We estimate that the standard time for revision of your journal, three months, will be enough to revise the manuscript, also including new experiments. We propose therefore to submit a point-to-point response to the requested revisions, alongside with the revised manuscript, asking to reconsider the manuscript as a revised one.

Thank you for your precious time in handling the manuscript so far and for the provided explanations.

Best regards
Paola Chiarugi

Professor of Biochemistry
Dept Experimental and Clinical Biomedical Sciences
School of Medicine, University of Florence
Viale Morgagni 50-50134
Florence

Dear Prof. Chiarugi,

Thank you for contacting us regarding the recent decision taken on your manuscript. As mentioned in the previous letter, should you be willing to address all concerns of the referees, we would be open to re-consider a revised manuscript. However, at this time we cannot promise that we can publish the study, given the fact that the revisions would have to include substantive experimentation to be compelling, and we can only move forward if the referees are convinced by the revision. For example, both referees #1 and #2 find that, as it stands, the data do not convincingly differentiate the contributions of tumor cell derived and CAF derived lactate, which is a concern directly related to the main message of the manuscript.

As per our editorial policy that allows a single round of major experimental revision, please note that the next decision will be the final one. I am looking forward to reading your revised manuscript when it is ready. Please find the guidelines for revision below my signature.

Kind regards,

Deniz Senyilmaz Tiebe

-
Deniz Senyilmaz Tiebe, PhD
Editor
EMBO Reports

We realize that it is difficult to revise to a specific deadline. In the interest of protecting the conceptual advance provided by the work, we recommend a revision within 3 months. Please discuss the revision progress ahead of this time with me if you require more time to complete the revisions, or if you have questions or comments regarding the revision (also by video chat).

1. A data availability section providing access to data deposited in public databases is missing (where applicable).
2. Your manuscript contains statistics and error bars based on $n=2$. Please use scatter plots in these cases.

You can submit the revision either as a Scientific Report or as a Research Article. For Scientific Reports, the revised manuscript can contain up to 5 main figures and 5 Expanded View figures, and it should not exceed 27000 characters. If the revision leads to a manuscript with more than 5 main figures it will be published as a Research Article. In this case the Results and Discussion section should be separate. If a Scientific Report is submitted, these sections have to be combined. This will help to shorten the manuscript text by eliminating some redundancy that is inevitable when discussing the same experiments twice. In either case, all materials and methods should be included in the main manuscript file.

4) a .docx formatted letter INCLUDING the reviewers' reports and your detailed point-by-point responses to their comments. As part of the EMBO publication's Transparent Editorial Process, EMBO reports publishes online a Review Process File (RPF) to accompany accepted manuscripts. This File will be published in conjunction with your paper and will include the referee reports, your point-by-point response and all pertinent correspondence relating to the manuscript.

<https://www.embopress.org/page/journal/14693178/authorguide#transparentprocess>

5) a complete author checklist, which you can download from our author guidelines

<https://www.embopress.org/page/journal/14693178/authorguide>. Please insert information in the checklist that is also reflected in the manuscript. The completed author checklist will also be part of the RPF.

6) Please note that all corresponding authors are required to supply an ORCID ID for their name upon submission of a revised manuscript (<<https://orcid.org/>>). Please find instructions on how to link your ORCID ID to your account in our manuscript tracking system in our Author guidelines

<<https://www.embopress.org/page/journal/14693178/authorguide#authorshipguidelines>>

7) Before submitting your revision, primary datasets produced in this study need to be deposited in an appropriate public database (see <https://www.embopress.org/page/journal/14693178/authorguide#datadeposition>). Please remember to provide a reviewer password if the datasets are not yet public. The accession numbers and database should be listed in a formal "Data Availability" section placed after Materials & Method (see also

<https://www.embopress.org/page/journal/14693178/authorguide#datadeposition>). Please note that the Data Availability Section is restricted to new primary data that are part of this study. * Note - All links should resolve to a page where the data can be accessed. *

Additional information on source data and instruction on how to label the files are available:

<https://www.embopress.org/page/journal/14693178/authorguide#sourcedata>

9) Our journal encourages inclusion of *data citations in the reference list* to directly cite datasets that were re-used and obtained from public databases. Data citations in the article text are distinct from normal bibliographical citations and should directly link to the database records from which the data can be accessed. In the main text, data citations are formatted as follows: "Data ref: Smith et al, 2001" or "Data ref: NCBI Sequence Read Archive PRJNA342805, 2017". In the Reference list, data citations must be labeled with "[DATASET]". A data reference must provide the database name, accession number/identifiers and a resolvable link to the landing page from which the data can be accessed at the end of the reference. Further instructions are available at <http://www.embopress.org/page/journal/14693178/authorguide#referencesformat>

10) Regarding data quantification (see Figure Legends:

<https://www.embopress.org/page/journal/14693178/authorguide#figureformat>)

- the name of the statistical test used to generate error bars and P values,

- the number (n) of independent experiments (please specify technical or biological replicates) underlying each data point,

- the nature of the bars and error bars (s.d., s.e.m.),

- If the data are obtained from n Program fragment delivered error ``Can't locate object method "less" via package "than" (perhaps you forgot to load "than"?) at //ejpvfs23/sites23b/embor_www/letters/embor_decision_revise_and_review.txt line 56.' 2, use scatter blots showing the individual data points.

12) Please also note our reference format:

Rebuttal Letter

Referee #1:

In this manuscript, Ippolito et al. study tumor-stroma interactions in prostate cancer (PCa) and propose that lactate secreted by cancer-associated fibroblasts (CAFs) promotes stem cell properties and invasiveness of PCa cells via a crosstalk involving collagen production and P4HA1-DDR1-STAT3 pathway. The authors show that CAF-conditioned medium and lactate activate secretion of collagen by PCa cells which leads to the acquisition of a pro-metastatic program. The idea that CAF-derived lactate promotes secretion of collagen implicated in PCa metastatic phenotype is novel and would be of interest to the cancer research community. However, the study suffers from a number of important issues and in places lacks rigor. Details below.

We thank the reviewer for appreciating our work. Below our replies to the arisen concerns.

Major Points:

1. The investigators propose CAFs as a source of secreted lactate in PCa leading to an interesting mechanistic insight into collagen production and modification to promote stem cell properties and invasion in PCa cells. Importantly, cancer cells can be a major source of lactate due to excessive glucose consumption and glycolysis (Warburg effect). Yet, this is not even mentioned by the authors. How significant are CAFs as a source of secreted lactate in tumors? Lactate levels from CAFs could simply be "a drop in the ocean" compared to other sources, such as cancer cells. This should be addressed. Adding conditioned medium from cultured CAFs to PCa cells does not establish CAFs as a significant lactate source within PCa tumors.

As previously reported by our group (), when CAFs are in contact with glycolytic cancer cells they undergo to a metabolic reprogramming leading to a OXPHOS rescheduling by lactate uploading. We described a lactate shuttle by which the CAFs derived lactate in the medium is suddenly uploaded by PCa cells and diverted to anabolic processes, TCA cycle and lipid droplets accumulation, driving an epigenetic response (PMID:22850421, PMID: 30936458, PMID: 35135811).

We agree with the reviewer's comment that lactate content within the tumor environment reasonably results from the contribution of different cellular sources, i.e. both CAFs and non-reprogrammed, glycolytic cancer cells. To better define this aspect, the impact of metabolic heterogeneity within the tumor mass and the contribution of all the metabolic sources of lactate has been emphasized in the revised version of the text (line 103). Importantly, in vitro analysis indicated that CAFs are one of the major contributors to lactate content in the tumor mass. Indeed, the quantification of lactate in conditioned media from CAFs and PCa cells reveals that prostate CAFs produce more lactate (~10 mM) than cancer cells (~3 mM) and healthy fibroblasts (HPF) (~3 mM). Moreover, lactate content in conditioned media from CAFs tends to decrease when administered to PCa cells, suggesting a tumor cell uptake, as previously reported by our group (PMID:22850421).

Even if these data suggest the contribution of stromal lactate over the tumoral one in affecting PCa cell behavior, we agree with the reviewer that the relative contribution of different cell populations in defining intratumoral lactate content has not been investigated. The reason for which we used extracellular lactate treatment is to highlight the impact of lactate environment *per se* (regardless of its origin) in prostate tumors, thus avoiding to restricting it to the CAF production only.

- Lactate extracellular levels in conditioned media from PCa cells, fibroblasts derived from HPF (from healthy areas within PCa tissues) and CAFs
- Lactate extracellular levels in conditioned media from CAFs and from PCa cells exposed to CAF-CM

2. The investigators show that lactate can promote collagen production in PCa cells. However, this is only shown *in vitro*. This should also be addressed *in vivo*. Is *col1a1* deposition increased in the mouse tumor models where CAFs are co-implanted, or mice are treated with lactate?

We thank the reviewer for noticing this limitation in our study. Therefore, we investigated Col1a1 expression in tumor masses derived from mice xenografts by IF and IHC analysis. Coherently, these approaches revealed higher content of Col1 in both CAF and lactate-treated tumor samples compared to controls, corroborating our *in vitro* data on Col1 deposition. We have included these data in the revised version of the manuscript as Appendix Figure 3b.

3. Figure 1e shows that CAF-CM and lactate can promote P4HA1 expression in PCa cells. However, whereas MCT1i leads to substantial P4HA1 repression in lactate samples, the same cannot be said about CAF-CM samples, which show a modest repression at best. This is essentially ignored by the authors but is important since the study proposes that stromal lactate enhances collagen synthesis by increasing P4HA1 expression via MCT1 transporter. Does this discordance undermine the idea that lactate from the CAF-CM is promoting P4HA1 in PCa cells? Could CAF-CM induction of P4HA1 be lactate independent?

We thank the reviewer for underlining this point and bringing out that the image could be misleading probably because of a bad resolution of the pdf file we provided during the process of submission. We therefore quantified the blot of CAF-CM samples (shown in Fig.1e) and proved that the repression on P4HA1 expression is quite strong following MCT1 inhibition. Also, by silencing MCT1 (EV Figure 1g), the effect on P4HA1 is clear, reinforcing our idea of stromal lactate inducing P4HA1 via lactate transport i.e. MCT1.

4. Based on Figure 2d and Supplementary Figure S3f, the authors state that "P4HA1 is required for an efficient survival in the circulation, extravasation, and retention in the secondary sites". However, it is not clear what mouse experiments in Figure 2d and Supplementary Figure S3f are really measuring. How was survival in circulation measured? What about extravasation? The investigators analyze the cell numbers in lungs 5 hours after intravenous injection. Is this enough time for extravasation? Were the cancer cells that were quantified in lungs, confirmed to be outside the vasculature in the lung parenchyma? There is also a major discordance between "representative" images and the quantifications. These experiments lack rigor.

We thank the reviewer for highlighting this point. We performed a lung retention assay to assess the ability of cancer cells to adhere and be retained from the lung (i.e. tumor cell arrest in the vasculature), as the first step in cancer cell colonization. The timing of the experiment has been set according to other publications exploring similar cancer cell phenotypes (PMID:24520024; 32015106). However, we agree with the reviewer that although P4HA1 induction in CAF- and lactate-treated PCa cells could impact survival in the circulation or lodging in the vasculature, the

extravasation step should be analyzed after 2-3 days (not 5 hours) from cancer cell injection, and this has not been done in our study. Considering all these aspects, we realized that we overestimated our results, and, for this reason, we now revised the manuscript to better clarify our approach and we provided a more precise data interpretation. Specifically, we removed the conclusions about cell survival in circulation and extravasation, and we now state: 'P4HA1 is required for an efficient retention in the lung, the first key step in the metastatic colonization' (lane 173). Finally, we apologize for the images provided in the previous version of the manuscript and we have replaced them with more representative ones.

5. In supplementary Figure 3a-b, there seems to be substantial experimental variations and lack of consistency. If experiments are repeated 3 times why would there be a 3-5 fold difference in invasion between control samples in 3A and 3B? This is surprising.

We apologize to the reviewer for this. This lack of consistency between two reported experiments is due to an error in counting the cells in the invasion assay experiments. To fix this, we have correctly repeated (and then replaced) the experiment shown in EV Fig.3b with a new one which is more coherent with the fold difference reported in the other figure.

6. In several figures, representative images of are not really representative when compared to quantifications. For example, in Figure 2b, images shown on the left do not represent the quantification on the right. The number of cells in CAF-CM and lactate condition with siP4HA1 cells appears to be much higher than the HPC-CM condition with siP4HA1 cells (not the case in the quantification). Also, in Figure 4d, images on the left show sphere sizes rather than sphere number which is quantified on the right.

We apologize that the figures provided for this experiment may be misunderstood, as the reviewer declares. However, the purpose of showing the pictures of prostaspheres is to provide an indication regarding the size of spheres rather than the number, which is reported effectively in the quantification plot. Indeed, sphere size is difficult to quantify by using imaging software but provides important information about the self-renewing ability of PCa cells. To clarify this point, we now added 'the size' to the results section.

7. Supplementary Figure 6c-d clearly shows that CAF-CM still phosphorylates pSTAT3 in P4HA1 knockdowns and cells treated with P4HA1 inhibitor, contrary to what the authors describe (lines 232-234). Please explain.

The reviewer points out a good question. Although our quantification shows decreased pSTAT3 levels in CAF-CM /P4HA1-impaired samples (now included in Appendix Fig.2c-d), we speculate that, as P4HA1 is an upstream molecule, its targeting may be more 'diluted' toward STAT3 phosphorylation, compared to other downstream target molecules (Col1, DDR1), that are more directly involved in STAT3 activation. Also, we know that CAF-CM may contain molecules other than lactate that are able to activate STAT3 (i.e. cytokines), leading to a residual activation of STAT3. To take account of this aspect, we performed the experiment incubating PCa cells with boiled CAF-CM to eliminate protein factors; interestingly, the induction of P4HA1 expression is maintained even in these experimental conditions, indicating the key contribution of metabolites (see WB image below). Moreover, we repeated the experiment with P4HA1 inhibitor DHB. pSTAT3 levels decrease in CAF-CM treated, and the pSTAT3 signal is comparable between boiled and not-boiled CAF-CM, supporting a metabolic regulation of the P4HA1/STAT3 loop. We provide results of these experiments below.

- WB analysis for P4HA1 in PCa cells exposed to HPF-CM, CAF-CM, boiled CAF-CM and lactate.
- WB analysis of pSTAT3 levels in PCa cells treated with HPF-CM, lactate, CAF-CM +/- P4HA1 inhibitor DHB and boiled CAF-CM. Quantification of pSTAT3/STAT3 was included as fold change numbers above the respective bands.

8. *The experiment presented in Figure 4b is very confusing. What concrete conclusions can be drawn from this? This should be explained thoroughly or removed.*

We apologize if these results seem unclear. Our goal was to assess whether lactate-reprogrammed cells differentially sense different collagen matrices produced by CAF or cancer cell by evaluating the phosphorylation of the collagen sensor DDR1. To better clarify this, we included a schematic picture describing the experimental setting we adopted (revised version of fig. 4b).

9. *In Figure 6D, it is unclear what is really metastasis in the lung images. Immunostaining of a PCa cell marker would provide the specificity required.*

To better clarify the presence of epithelial cancer cells in the lung metastasis, we performed IF by using an E-cadherin antibody as an epithelial PCa cell marker, and we observed a specific staining in lung tissues, indicating that PCa cells have reached the metastatic site. We have included these new figures in the revised form of the manuscript as Appendix Fig. 3c.

10. Quantification is missing in several immunofluorescence analyses (Figure 1c, S1c-d, 5f, S7e).

We have included the quantification plots along the images.

11. The quality of western blots is poor in several panels, such as Figure 3b, Figure 5c and 5g.

We have replaced the poor-quality blots in Figure 3b and 5g with better ones.

Minor Points:

1. *For the table presented at the end of the manuscript, a table legend and further explanations in the text are needed.*

We thank the reviewer for the suggestion. We have included a table legend along the Figure 1 legend and we inserted an descriptive sentence (line 114).

2. *In Figure 1c, it is not clear how long the PCa cells were exposed to HPF- CAF-CM or lactate. This is neither mentioned in the text nor the Methods. The same applies for all results obtained with conditioned medium or lactate supplementation.*

We usually expose cancer cells to CM or lactate for 48 hours, as stated in line 110. We further mention this point in the Methods section (line 391).

3. *In Figure 3a, a trendline is missing.*

The data from the Fig.3a are in silico data, that derived from the GEPIA (<http://gepia.cancer-pku.cn>) webtool, which unfortunately do not insert the trendline in the correlation plots, giving the Pearson and *p* values only.

4. *A model to visualize all the mechanistic players involved in the proposed findings, would be helpful to crystalize the take home message of the paper.*

Thanks for this helpful and clarifying suggestion. A graphical scheme has been previously uploaded separately and thus you can find it as separate file.

5. *Figure labeling mix-ups.*

line 193 : Figure 3g (not 3f)

line 196 : Figure 3h (not 3g)

Figure 1 legend : panels g and h are inversed. Moreover, panel h is wrongly termed f in the legend).

Supp Fig 1 legend : panels c, d and e completely mixed.

Supp Fig 3f legend : n=3 mice per group mentioned whereas n=4 in the Methods.

Supp Fig 4 and legend : panels not in alphabetical order.

Supp Fig7 : Last panel is f not g.

We made the corrections accordingly.

Referee #2:

Ippolito and colleagues propose that CAF-secreted lactate promotes the expression of prolyl-4-hydroxylase (P4HA1) in PCa cells, which in turn leads to de novo collagen-induced DDR1 activation and increased invasiveness. The work is elegantly carried out but lacks experimental data to make the context of the study more biologically relevant.

We thank the reviewer for appreciating our work. Below our replies.

1) The paradigm of lactate-generating CAF developed in the current manuscript is different from data published by others where high lactate concentrations in tumors are used by CAF to support progression instead of CAF generating lactate which in turn influences the phenotype of cancer cells.

See for instance: <https://www.ncbi.nlm.nih.gov/pmc/articles/PMC8606002/>

The CAF contribution to tumor lactate vs. the production of lactate by cancer cells is all the more unclear as hypoxic prostate cancer cells are largely glycolytic and locally produce high lactate concentrations in tumors. The authors themselves refer to the control of P4HA1 by the hypoxia-inducible transcription factor (HIF-1).

We thank the reviewer for discussing this point. We agree with him/her that cell types in the tumor microenvironment utilizing lactate is a raising question to be addressed. However, we think the point may be framed in a tissue-dependent manner. Indeed, tumor-associated fibroblasts have been reported to secrete lactate specifically in certain types of solid tumors, including prostate and breast carcinomas (PMID: 32492417; 25732824; 22850421), thus contributing to lactate accumulation in these tumor models. Differently, CAF exploiting tumor-derived lactate have been observed in pancreatic tumor models (PMID: 37733442; 34764457). This discrepancy suggests the existence of a tumor-type specific metabolic adaptation in CAFs. According to this interpretation, we have preliminary data indicating that pancreatic cancer cells are refractory to the metabolic rewiring following exogenous lactate administration, e.g. lipid anabolism as previously addressed in our published work on prostate cancer cells (PMID: 35135811). This may be due to the massive fermentative metabolism of pancreatic cancer cells, compared to other tumor types, such as PCa.

Hence, in different tumors the lactate shuttle may have different directions, always engaging tumor and fibroblast cells.

Concerning HIF1 regulation, we have reported a pseudo-hypoxic stabilization upon lactate entry in PCa cells, in keeping with previous reports from the Sonveaux laboratory (PMID: 21235356, PMID: 23082126). Hence, lactate entry can boost HIF1 expression under normoxic conditions, and this may be relevant for P4HA1 induction in our cells.

2) *The use of CM may look straightforward but introduces bias. First, the extent of lactate concentration in CM from CAF and HPF is not indicated. Second, if differences result from a higher glycolytic turnover in CAF (than in HPF), there are several other metabolic intermediates that are different between the two CM. To which extent reduced glucose and increased (extracellular) pyruvate concentrations influence the phenotype of treated cells is unclear. Same comment for the buffering capacity of the medium which may be altered in CAF medium due to H⁺ release together with lactate.*

Also, does addition of the CAF-CM amount of lactate in HPF-CM recapitulate the reported effects? Does the acidification of HPC-CM and reduced glucose concentration in HPC-CM fail to mimic these effects?

We thank the reviewer for highlighting these concerns. We previously published that lactate content in CAF-derived CM is higher than HPF counterpart (PMID: 22850421). To better clarify this aspect, we also measured lactate levels in fibroblasts extracted from human PCa specimens, and we found the lactate levels in CAFs are around double compared with the fibroblasts from healthy prostatic tissues (BPH: benign hyperplasia). In in vitro context, CAFs produce ~10 mM lactate, compared to fibroblasts derived from healthy areas within PCa (HPFs, 3-4mM), thus exhibiting a similar trend with the in vivo analysis. By investigating the CM content of the major metabolic released components, we observed that lactate is the most altered comparing the two samples. Importantly, no differences were observed in pyruvate levels (note that pyruvate levels in the medium – 1mM – are supraphysiological, limiting its use to above concentrations). To investigate the real contribution of lactate in driving the effect observed following CAF-CM incubation, we approached HPF-CM by i) adding 6 mM lactate to mimic CAF-CM, ii) acidifying it by adding 1N HCl to adjust pH at 6.5 and iii) collecting following 48h of incubation in low-glucose media. We found that the sole addition of lactate to HPF-CM supports P4HA1 and Col1 expression, while no effects were observed in response to other conditions. These results support the key role of lactate in the described phenotype.

- Lactate extracellular levels in fibroblasts derived from BPH (benign prostatic hyperplasia) and CAFs through GC-MS analysis.
- Lactate extracellular levels in conditioned media from PCa cells, fibroblasts derived from HPF (from healthy areas within PCa tissues) and CAFs
- Heatmap depicting extracellular levels of the main representative environmental metabolites such as pyruvate, lactate and glutamine
- WB blot in PCa cell treated with CAF-CM, lactate, acidified medium (HCl), and HPF-CM supplemented with 6mM Lactate or collected after HPF culturing in low-glucose medium
- IF representative images for Col1 of PCa cells treated as indicated.

3) *The use of lactic acid (as indicated in Methods) instead of lactate (indicated in Results) may account for some differences related to acidification. This should be addressed. Same for the concentrations used. Many papers in the field relate to a concentration of 20-40mM because of a wrong interpretation of old papers expressing the concentrations in mg/ml. The rationale for the lactate concentration should be supported by experimental data validating the chosen concentration for in vitro studies.*

We thank the reviewer for specifying this point. We know acidification is very important in cancer dynamics, and we did not underestimate the problem. However, we used buffered media for culturing cells and adding lactic acid (and the consequent drop in pH) is immediately counteracted in 1-2 minutes. Also, buffer acidification does not promote P4HA1/Col1 induction, as shown in the above IF image. Furthermore, we used lactic acid, not sodium lactate because its transport is mediated by MCT1, which acts as H⁺/lactate co-transporter (PMID: 21930917). To avoid further misinterpretation, we used the term lactic acid (LA) in the revised text and figures.

Although the most widely used in vitro lactate concentration is 10 mM, other studies reported 20 mM (PMID: 30563827; 33296645; 37863066; 24928781). In addition, we measured lactate content in tumor xenografts generated by subcutaneously injecting both CAFs and PCa cells, and we found lactate levels range around 40-50 μmol/grams within the tumor masses. This is in keeping with seminal papers (PMID: 15279558; Lactate and Acidity in the Cancer Microenvironment, Annual Review of Cancer Biology, 2020) stating as follows: "Lactate concentrations ranged across tumors from 0 to 50 μmol/g of tissue, which corresponds to approximately 0–50 mM in a liquid phase". Thus, we set 20 mM as the upper limit for our in vitro treatment for two reasons: i) higher concentrations above 30 mM negatively affect cell viability in in vitro settings; ii) it resembles the heterogenous lactate environment derived from the levels produced in vitro by the main PCa cell populations (CAF, HPF and PCa cells).

Therefore, these indications led us to choose 20 mM lactate in vitro to better recapitulate the PCa 'lactagenic' environment and to maximize the effect of such metabolite on tumor cell behavior.

Specifics:

- Lack of changes in cell proliferation and in vitro survival upon P4H1, DDR1, COL1A1 and STAT3 silencing should be documented.

No substantial changes in proliferation rate were observed in PCa cells silenced for the abovementioned target genes, as shown in the figures below.

- Cell proliferation assay assessed by counting PCa cells, treated or not (NT) with lactate, previously transfected with siCTR or the respective targeting siRNAs as indicated, for 24, 48 and 72 hours.

- Stimulatory effect of lactate on total STAT3 is visible in different panels of fig 5. The balance between changes in phosphorylated form and total abundance should be clarified.

We thank the reviewer for having noticed this point. Even if we did not investigate in depth the molecular basis of this phenomenon as it is not the scope of the study, we know that, in some context, the stability of STAT3 could be important beyond its phosphorylated form. This evidence has been shown in recent papers linking higher total STAT3 abundance to a higher tumor aggressiveness in prostate cancer cells (PMID: 31474764; 18483213) (e.g. apoptosis resistance and migratory capacity). These data likely corroborate the aggressive traits we observed in our experimental models.

- Injection (i.p) of lactate in mice may lead to systemic effects making interpretation of data confusing. The impact of serum lactate on tumor lactate concentration should be proved.

We thank the reviewer for bringing this proper consideration of in vivo lactate treatment. The only route of administration of lactate in vivo is i.p. injection, and we set our i.p. injection according to a study (PMID: 27218871) providing evidence that blood lactate concentration reaches a spike of 18 mM after the injection i.e. 1g/kg lactate and returns to a baseline 1-2 mM after few hours. This dose was chosen to approximate the levels and persistence of rises in plasma lactate that occur following an established tumor-associated environment and circulating levels, as observed in other models (PMID: 36615018). Also, a couple of papers (PMID: 35950184; 37055619) described that chronic lactate administration leads to high serum lactate levels with no significant systemic effects regarding energy expenditure, mice body weight, etc. Finally, the increased Col1 deposition in these samples analyzed by IHC clear proof that lactate administration has effects on the tumor tissues in vivo, as expected (Appendix Fig. 3b).

Minor:

- Fig 1C : contrast should be optimized to better see fluorescent signals

A better optimized pictures were included in Fig.1C.

Referee #3:

Ippolito et al. provide novel metabolic insights in the metastatic behavior of prostate cancer. The authors uncovered a new mechanism in prostate cancer (PCa) cells, by which cancer-associated fibroblasts (CAF)-secreted lactate promotes expression of genes coding for the collagen family. In detail, lactate-utilizing PCa cells rely on α -ketoglutarate (α -KG), which activates the α -KG-dependent collagen prolyl-4-hydroxylase (P4HA1) to support collagen hydroxylation. In turn, de novo synthesized collagen exerts a signaling role by activating discoidin domain receptor 1 (DDR1), supporting stem-like and invasive features of PCa cells.

The authors have shown nicely that the inhibition of lactate-induced collagen hydroxylation and DDR1 activation reduces metastatic colonization in vitro and in vivo. Interestingly, STAT3 activation is crucial to control the invasive and stem-like features of lactate-rewired PCa cells. Furthermore, the authors provide evidence that the lactate-dependent P4HA1/Col1/DDR1 axis is critical for STAT3 activation.

Overall, this is an important manuscript providing a new understanding of the P4HA1/Col1/DDR1 signaling axis regulating prostate cancer metastasis and link collagen remodeling/signaling and the nutrient environment exploited by PCa cells. The authors present their findings clearly and describe them accurately within the text. I really enjoyed reading this paper. A real strength of this study is the identification of a unique target to block metastasis and that the paper connected the metabolic and molecular consequences and presented novel therapeutic approach.

We thank the reviewer for his/her careful appraisal of our manuscript.

Below are comments that aim to further increase the clarity and significance of this study:

1. Does the expression of MCT1 increase when DU145 cells are exposed to CAF-CM or lactate? Can the authors evaluate the efficiency of MCT1 inhibition in tumor cells?

As shown in Fig.EV1e, MCT1 expression in tumor cells increases upon CAF-CM and lactate exposure. Also, MCT1 inhibitor administration leads to ~40% effective inhibition of lactate uptake, as assessed by a radioactive ¹⁴C-Lactate uptake assay in CAF-CM-treated tumor cells.

- ¹⁴C-Lactate uptake assessed in PCa cells treated with CAF-CM and CAF-CM + MCT1 inhibitor for 48h.

2. Can the authors comment whether the pH of the medium was adjusted when supplemented with 20 mmol/L lactate.

The importance of pH is not underestimated in our model. However, using buffered media, the supplementation of 20mM lactate does not greatly affect the medium pH during the cell treatment, as pH drop is quickly neutralized after 1-2 minutes. For this reason, we do not need to adjust the pH of the medium. Also, we acidified the medium by addition of 1N HCl and we do not see a substantial difference in P4HA1 expression, suggesting that the only addition of the metabolite lactate is effective in inducing P4HA1 (WB image below).

- WB analysis of P4HA1 in PCa cells exposed to HPF-, CAF-CM, lactate or acidified medium (HCl).

3. Do P4HA1-silenced PCa cells metastasize to other organs other than the lung (Fig. 2D)

We appreciated the question about the in vivo metastatic behavior of prostate cancer cells. Nevertheless, to our knowledge DU145 cell line has not been documented to metastasize to major organs except the lungs in mice after subcutaneous xenograft injection.

4. Can the authors speculate/discuss how lactate conditioning promotes DDR1 expression?

We appreciated the reviewer's comment. A paper described that DDR1 expression is directly induced by collagen (PMID: 21335558). This may be true even in our case. Still, we speculate that lactate-induced collagen promotes DDR1 activation, and the downstream signalling node STAT3 activation could be responsible for feeding DDR1 expression, as STAT3 is found on DDR1 promoter gene.

5. The authors displayed nicely that STAT3 inhibition negatively impacts CAF/lactate enhanced invasiveness and prostaspheres formation. Can the authors provide evidence that STAT3 overexpression impacts CAF/lactate invasiveness? Does STAT3 overexpression lead to a reversed effect on invasiveness and prostaspheres formation?

The reviewer is referring to STAT3 role in lactate-induced invasiveness/spheres formation. However, STAT3 expression is not promoted by lactate. For this reason, STAT3 overexpression should be ineffective to promote aggressive traits of PCa cells, neither to have a reversed effect.

6. The authors displayed representative western blot images, indicating that lactate/driven DDR1 and Col1 expression are significantly reduced in STAT3-silenced or STAT3-inactive PCa cells exposed to stromal lactate. However, representative western blot images do not give statistical

information about changes in the protein abundance. In order to provide evidence on significant changes in the protein abundance, could the authors add a statistical western blot analysis to the respective figure or in the supplemental part?

We have included a numerical quantification on the respective figure.

7. Could the authors give more information about the RNA sequencing method? How many reads per samples were measured? What reference genome was used in order for subsequent alignment and which software was used? Did the authors correct the resulting p-values for multiple testing?

Here the information about our RNA-seq experiment.

RNA-seq for all experiments was performed at the Institute of Oncology Research using Next Ultra II Directional RNA Library Prep Kit for Illumina starting from 800 ng of total RNA each sample and sequenced on the Illumina NextSeq500 with single-end, 75 bp long reads.

Sample	# of sequenced reads	# of aligned reads
Ctrl_1	19835460	15867437
Ctrl_2	16633050	13300345
Ctrl_3	19110945	15207422
Ctrl_4	17603402	14030194
LA_2	17915331	14243892
LA_3	21285002	17173584
LA_4	16578836	12999837

Reads were aligned to the reference genome using STAR 2.6.1 along with the release of human genome assembly (GRCh38.p12, v29). Computed p-values were adjusted for multiple testing using the Benjamini-Hochberg (FDR) correction method.

8. It would be helpful to add a graphical abstract or a final figure in order to summarize the novel identified P4HA1/Col1/DDR1 signaling axis that regulates prostate cancer metastasis.

Thanks for this helpful and clarifying suggestion. A graphical scheme has been previously uploaded separately and thus you can find it as separate file.

Minor comments:

Line 174: Reference is stated as PMID and should be edited.

Done.

Line 193 (Figure 3f): Figure 3f displays ALDH1A1 expression, not the ability to intra/extravasate sustained by lactate

Revised as requested.

Line 195 (Figure 3g): Figure 3g shows fluorescently labeled DH145 cells, not DDR1 expression in Gleason-graded PCa patient samples. Figure 3h is not described in the results section.

Revised as requested.

Line 259: There is no supplemental Figure S7f

Revised as requested.

Line 401: Could the authors add the sequences of all used siRNAs?

Unfortunately, siRNAs sequences are not provided by the company.

Dear Dr. Deniz Senyilmaz Tiebe,

thank you for your kind reply and for forwarding an overall positive revision of our manuscript.

We greatly appreciate your proposal to discuss possible improvements and the experimental plan for addressing the remaining concerns of Ref1. Please find below our proposed experimental approach:

POINT TO POINT ANSWERS TO REF1:

The authors have revised and somewhat improved the manuscript. For example, they demonstrate (argue with references) that CAFs are a major source of lactate in PCa, as compared to the PCa cells themselves. They also show increased expression of Collagen Type I in CAF or lactate treated xenograft tumors and make significant other revisions. However, I think there are a few issues that are still unresolved.

We thank the reviewer for appreciating the improvements of the revised manuscript and we will solve the remaining issues.

Specific points:

1. My main criticism is regarding the claimed metastatic nodules in Figure 6d and Appendix Fig. 3c. First, the quality of Figure 6d has not been improved at all. As someone who has been studying lung metastasis for many years, I would not be confident that there is a single real metastasis in these panels (Scr with CAF or Lactate data are definitely not convincing). Moreover, although the IF staining with E-cadherin antibody may appear to show metastatic nodules, the staining is very strange and does not show results that I would be confident with. The E-cadherin antibody used in the IF staining is #3195 from Cell Signaling Technology. This antibody is not human specific and also reacts to mouse E-cadherin. Therefore, we should expect substantial staining in lung epithelial cells. However, this is not the case when looking at the images in Appendix Fig. 3c. I do not have confidence in these results. First, E-cadherin is not a good marker here. Second, the presented images are really strange.

We are aware that metastatic lesions in our lung IHC images are small, but unfortunately, the prostate cancer xenograft model is known to promote only micro-metastasis formation and not large metastatic nodules. The point of the Referee about the antibody specificity is well taken. However, in our experimental setting, we have limited the interference of the mouse tissues by decreasing the fluorescence gain in our confocal analysis. Nevertheless, as correctly pointed out, we cannot exclude cross-reactivity. We could therefore validate the presence of human (tumor) cells in the lung tissues with one of the following approaches: (1) repeat the analysis on our paraffin embedded tissues, carrying out IHC/IF staining of lung sections by using a human-specific pan-Cytokeratin (ab234297, Abcam) and a human specific E-cadherin (ab40772, Abcam) antibody; (2) if the quality of the images would not be satisfactory and leave some doubts about the presence of human cell within the lung, we could alternatively, detect human Alu sequences by digital droplet PCR (ddPCR). A ddPCR based approach will indeed detect the presence of human DNA in mouse lungs, although this method will not help in quantifying human cells or visualizing metastases.

2. The investigators show that CAF-CM contains higher lactate than PCa-CM and HPF-CM. If the presented data is a unique repeat and not a direct copy from the cited publications, I think it would be worth including in this manuscript to support the context.

These data have been obtained specifically for the revision, but very similar results have been already published in our previous works (PMID: 22850421, Can Res 2012). As these new data have been obtained in different CAFs lineages, obtained from different patients with respect to those published in 2012, we can include these in a new supplementary figure. In addition, concerning data published in 2012, we have some new data to include in the new supplementary figure, such as intratumoral lactate quantification, and the effect of MCT1 inhibitor on lactate uptake (suggested by reviewer#3).

3. I can't say that I am impressed with investigators argument for my point nr.3. Even considering the quantification, CAF-CM induction of P4HA1 is not as well neutralized by MCTi compared to LA induced P4HA1 in Figure 1e. The knockdown data in EV Figure 1g may be more convincing, however, the P4HA1 western blot is of poor quality, and doesn't look publication quality.

We understand this point. One possible explanation is that the complexity of CAF-CM, containing a mixture of both metabolites and cytokines secreted by CAFs, could be accounted for this incomplete P4HA1 expression abrogation. To investigate this aspect, we propose to repeat the experiment by introducing a CAF-CM boiled sample \pm MCT1 inhibitor. A boiled CM will allow to eliminate the thermolabile molecules within the CM, including cytokines and chemokines. This approach will let emerge the role of metabolites, including lactate in the CAF-CM. Moreover, we will replace the EV Fig.1g with a better quality one.

4. The response to my point nr. 7 on STAT3 phosphorylation is not convincing. The examples of western blots shown in response to reviewers are not impressive. Personally, I would have problem trusting this data to show any real differences.

We apologize for the low-quality image, likely due to the use of an old immunoblot acquisition instrument, due to a temporary breakdown of the device that we routinely use. The breakdown has been solved and we can repeat the experiment, again by introducing a CAF-CM boiled sample \pm DHB and/or P4HA1 silencing.

5. The cartoon shown in Figure 4 to explain western blot data only shows half the experiment. It only shows the generation of cell free matrix, not what is plated on top and what is measured by western blot. I also think that the text needs to be revised to really clarify what point the authors want to make here.

We will modify and revise the cartoon and the text accordingly, taking into consideration the Referee's concerns.

6. In the response to my point nr. 6 on PCa spheres, the authors mention that sphere size is difficult to quantify using imaging software. I disagree, it should not be difficult to quantify sphere sizes. However, this is not a major point, so I leave it to the authors' discretion.

We will quantify the size of spheres using a new software, since the pipeline for spheres quantification has now been optimized in the lab.

Responses to Reviewer 2

The responses to Reviewer 2 seem to be fine overall. However, in the response to question nr. 2, the authors show experimental results (for example on lactate levels in different conditioned media as well as comparison with pyruvate and glutamine levels), but do not include these results in the paper. I would think the results would be helpful addition to the paper, unless these exact plots are directly lifted from previous publications. The authors also provide further data in response to question nr. 3, such as intra-tumoral lactate levels to justify the concentration that they use for cell treatment in vitro. I would think these data would also be helpful to include in the manuscript (unless the exact plot is already published).

We thank the reviewer for having appreciated the revised manuscript. Also, we agree with the suggestion to include new experimental results, and we include intra-tumoral lactate levels, to better justify in vitro treatment with lactate we used (see Point 2 of Ref 1).

Referee #3:

This is a revised version of the previously submitted manuscript from Ippolito and colleagues. The authors have performed a detailed revision of the manuscript and included novel data. The authors have fully addressed all of my comments.

Thank you for having appreciated our response,

My best regards

Paola Chiarugi

Dear Dr. Chiarugi,

Thank you for submitting your revised manuscript. It was now seen by two of the original referees, whose reports are included below. Referee #1 also assessed the response to the concerns of referee #2, who was unavailable for re-reviewing the revised version.

I apologize for this unusual delay in getting back to you. It took longer than anticipated to receive the referee reports given the recent busy time of the year.

Referees acknowledge that the revised manuscript is notably improved. However, referee #1 has significant outstanding concerns that need to be addressed to consider publication here. In particular, the referee does not find the stainings of the metastatic nodules convincing (point 1). Moreover, referee does not find the blots depicting P4HA1 levels (point 3) and STAT3 phosphorylation (point 4) of sufficient quality. At this stage, I find it most constructive to ask upfront for a point-by-point response outlining what you can do to address the raised concerns. You might already have good arguments/data at hand regarding these points. Based upon this I will then be able to take the decision.

Please contact me before you embark upon new experiments. I'm looking forward to your response.

Yours sincerely,

Deniz Senyilmaz Tiebe

Deniz Senyilmaz Tiebe, PhD
Scientific Editor
EMBO Reports

Referee #1:

The authors have revised and somewhat improved the manuscript. For example, they demonstrate (argue with references) that CAFs are a major source of lactate in PCa, as compared to the PCa cells themselves. They also show increased expression of Collagen Type I in CAF or lactate treated xenograft tumors and make significant other revisions. However, I think there are a few issues that are still unresolved.

Specific points:

1. My main criticism is regarding the claimed metastatic nodules in Figure 6d and Appendix Fig. 3c. First, the quality of Figure 6d has not been improved at all. As someone who has been studying lung metastasis for many years, I would not be confident that there is a single real metastasis in these panels (Scr with CAF or Lactate data are definitely not convincing). Moreover, although the IF staining with E-cadherin antibody may appear to show metastatic nodules, the staining is very strange and does not show results that I would be confident with. The E-cadherin antibody used in the IF staining is #3195 from Cell Signaling Technology. This antibody is not human specific and also reacts to mouse E-cadherin. Therefore, we should expect substantial staining in lung epithelial cells. However, this is not the case when looking at the images in Appendix Fig. 3c. I do not have confidence in these results. First, E-cadherin is not a good marker here. Second, the presented images are really strange.
2. The investigators show that CAF-CM contains higher lactate than PCa-CM and HPF-CM. If the presented data is a unique repeat and not a direct copy from the cited publications, I think it would be worth including in this manuscript to support the context.
3. I can't say that I am impressed with investigators argument for my point nr. 3. Even considering the quantification, CAF-CM induction of P4HA1 is not as well neutralized by MCTi compared to LA induced P4HA1 in Figure 1e. The knockdown data in EV Figure 1g may be more convincing, however, the P4HA1 western blot is of poor quality, and doesn't look publication quality.
4. The response to my point nr. 7 on STAT3 phosphorylation is not convincing. The examples of western blots shown in response to reviewers are not impressive. Personally, I would have problem trusting this data to show any real differences.
5. The cartoon shown in Figure 4 to explain western blot data only shows half the experiment. It only shows the generation of cell free matrix, not what is plated on top and what is measured by wester blot. I also think that the text needs to be revised to really clarify what point the authors want to make here.
6. In the response to my point nr. 6 on PCa spheres, the authors mention that sphere size is difficult to quantify using imaging software. I disagree, it should not be difficult to quantify sphere sizes. However, this is not a major point, so I leave it to the authors' discretion.

Responses to Reviewer 2

The responses to Reviewer 2 seem to be fine overall. However, in the response to question nr. 2, the authors show experimental results (for example on lactate levels in different conditioned media as well as comparison with pyruvate and glutamine levels), but do not include these results in the paper. I would think the results would be helpful addition to the paper, unless these exact plots are directly lifted from previous publications. The authors also provide further data in response to question nr. 3, such as intra-tumoral lactate levels to justify the concentration that they use for cell treatment in vitro. I would think these data would also be helpful to include in the manuscript (unless the exact plot is already published).

Referee #3:

This is a revised version of the previously submitted manuscript from Ippolito and colleagues. The authors have performed a detailed revision of the manuscript and included novel data.

The authors have fully addressed all of my comments.

Dear Prof. Chiarugi,

Thank you for submitting your preliminary point-by-point response. I have now looked at your points carefully. I appreciate that you can address the concerns raised and see that the data proposed to be added will strengthen the manuscript.

Moreover, I need you to address the editorial points below.

- The Data Availability section, which is currently named as Data and code availability, is reserved for the primary datasets generated in the study. I note that the dataset GSE195639 was already published as a part of your 2022 study (PMID: 35135811). Please remove this section from the manuscript, and cite GSE195639 in the form of data citation in the text where appropriate. Please confirm that Figures 1a and b, which are derived from GSE195639, were not previously published in other publications. Please see <https://www.embopress.org/page/journal/14693178/authorguide#referencesformat> for examples of data citation.
- Please rename the 'Authors' Disclosures' section as 'Disclosure Statement and Competing Interests' and place it after Acknowledgements.
- Please remove the Authors' Contributions section from the manuscript.
- Please make sure that the funding information is complete in the manuscript submission system.
- Please resubmit Table EV1 as Dataset EV1. The file needs to be renamed as Dataset EV1 and the callout in the manuscript needs to be updated accordingly. Its legend needs to be removed from the manuscript text and included in the same Excel file (as a separate tab/sheet).
- The Appendix file needs a Table of Contents with page numbers. The figures should be renamed as Appendix Figure S1, etc.
- Please rearrange the source data as one (zip) file per figure - i.e. please include the contents of the folder named 'uncropped WB Source Data' into the corresponding figure files.
- During our routine figure checks, we noted a potential re-use of image between Figure 4D and Figure EV4 G (HPF-CM / siCOL1A1). Please clarify.
- Our production/data editors have asked you to clarify several points in the figure legends:
 - o Please note that a separate 'Data Information' section is required in the legends of figures 1c-d, f; 2a-d; 3c-h; 4c-d; 5d-f; 6c, e; EV 1b-d; EV 2b, e; EV 4e-f, l-m; EV 5a-d.
 - o Please indicate the statistical test used for data analysis in the legends of figures 1b; EV 4e-f, l-m; EV 5b-d.
 - o Please note that in figures 1c-d, f; 2d; 3c-h; 5e-f; EV 3a, f; EV 4a, e-f; there is a mismatch between the annotated p values in the figure legend and the annotated p values in the figure file that should be corrected.
 - o Please note that the box plots need to be defined in terms of minima, maxima, centre, bounds of box and whiskers, and percentile in the legends of figures 1c; 2d; 5f; 6c; EV 1d; EV 3f.
 - o Please note that information related to n is missing in the legends of figures 1c, f; 6f-g; EV 2b; EV 4e-f, l-m; EV 5b-d.
 - o Please note that the scale bar is missing for figures 2b, d; 3g; 4d; 5e; EV 1c; EV 3e.
- Papers published in EMBO Reports include a 'synopsis' and 'bullet points' to further enhance discoverability. Both are displayed on the html version of the paper and are freely accessible to all readers. The synopsis includes a short standfirst summarizing the study in 1 or 2 sentences (max 35 words) that summarize the paper and are provided by the authors and streamlined by the handling editor. I would therefore ask you to include your synopsis blurb and 3-5 bullet points listing the key experimental findings.
- The synopsis image needs to be 550px wide and 300-600px high. When your synopsis image is resized accordingly, some of the labels are too small to read (please see attached). Please provide a synopsis image with larger labels.

Thank you again for giving us to consider your manuscript for EMBO Reports, I look forward to your minor revision.

Kind regards,

Deniz Senyilmaz Tiebe

--

Deniz Senyilmaz Tiebe, PhD
Editor
EMBO Reports

Referee #1:

The authors have revised and somewhat improved the manuscript. For example, they demonstrate (argue with references) that CAFs are a major source of lactate in PCa, as compared to the PCa cells themselves. They also show increased expression of Collagen Type I in CAF or lactate treated xenograft tumors and make significant other revisions. However, I think there are a few issues that are still unresolved.

Specific points:

1. My main criticism is regarding the claimed metastatic nodules in Figure 6d and Appendix Fig. 3c. First, the quality of Figure 6d has not been improved at all. As someone who has been studying lung metastasis for many years, I would not be confident that there is a single real metastasis in these panels (Scr with CAF or Lactate data are definitely not convincing). Moreover, although the IF staining with E-cadherin antibody may appear to show metastatic nodules, the staining is very strange and does not show results that I would be confident with. The E-cadherin antibody used in the IF staining is #3195 from Cell Signaling Technology. This antibody is not human specific and also reacts to mouse E-cadherin. Therefore, we should expect substantial staining in lung epithelial cells. However, this is not the case when looking at the images in Appendix Fig. 3c. I do not have confidence in these results. First, E-cadherin is not a good marker here. Second, the presented images are really strange.
2. The investigators show that CAF-CM contains higher lactate than PCa-CM and HPF-CM. If the presented data is a unique repeat and not a direct copy from the cited publications, I think it would be worth including in this manuscript to support the context.
3. I can't say that I am impressed with investigators argument for my point nr. 3. Even considering the quantification, CAF-CM induction of P4HA1 is not as well neutralized by MCTi compared to LA induced P4HA1 in Figure 1e. The knockdown data in EV Figure 1g may be more convincing, however, the P4HA1 western blot is of poor quality, and doesn't look publication quality.
4. The response to my point nr. 7 on STAT3 phosphorylation is not convincing. The examples of western blots shown in response to reviewers are not impressive. Personally, I would have problem trusting this data to show any real differences.
5. The cartoon shown in Figure 4 to explain western blot data only shows half the experiment. It only shows the generation of cell free matrix, not what is plated on top and what is measured by western blot. I also think that the text needs to be revised to really clarify what point the authors want to make here.
6. In the response to my point nr. 6 on PCa spheres, the authors mention that sphere size is difficult to quantify using imaging software. I disagree, it should not be difficult to quantify sphere sizes. However, this is not a major point, so I leave it to the authors' discretion.

Responses to Reviewer 2

The responses to Reviewer 2 seem to be fine overall. However, in the response to question nr. 2, the authors show experimental results (for example on lactate levels in different conditioned media as well as comparison with pyruvate and glutamine levels), but do not include these results in the paper. I would think the results would be helpful addition to the paper, unless these exact plots are directly lifted from previous publications. The authors also provide further data in response to question nr. 3, such as intra-tumoral lactate levels to justify the concentration that they use for cell treatment in vitro. I would think these data would also be helpful to include in the manuscript (unless the exact plot is already published).

Referee #3:

This is a revised version of the previously submitted manuscript from Ippolito and colleagues. The authors have performed a detailed revision of the manuscript and included novel data.

The authors have fully addressed all of my comments.

POINT-TO POINT EMBOR-2023-57799V4

We have taken up the text of the pre-decision consultation. Our answers are highlighted in **bold**.

POINT TO POINT ANSWERS TO REF1:

The authors have revised and somewhat improved the manuscript. For example, they demonstrate (argue with references) that CAFs are a major source of lactate in PCa, as compared to the PCa cells themselves. They also show increased expression of Collagen Type I in CAF or lactate treated xenograft tumors and make significant other revisions. However, I think there are a few issues that are still unresolved.

We thank the reviewer for appreciating the improvements of the revised manuscript and we will solve the remaining issues.

Specific points:

1. My main criticism is regarding the claimed metastatic nodules in Figure 6d and Appendix Fig. 3c. First, the quality of Figure 6d has not been improved at all. As someone who has been studying lung metastasis for many years, I would not be confident that there is a single real metastasis in these panels (Scr with CAF or Lactate data are definitely not convincing). Moreover, although the IF staining with E-cadherin antibody may appear to show metastatic nodules, the staining is very strange and does not show results that I would be confident with. The E-cadherin antibody used in the IF staining is #3195 from Cell Signaling Technology. This antibody is not human specific and also reacts to mouse E-cadherin. Therefore, we should expect substantial staining in lung epithelial cells. However, this is not the case when looking at the images in Appendix Fig. 3c. I do not have confidence in these results. First, E-cadherin is not a good marker here. Second, the presented images are really strange.

We are aware that metastatic lesions in our lung IHC images are small, but unfortunately, the prostate cancer xenograft model is known to promote only micro-metastasis formation and not large metastatic nodules. The point of the Referee about the antibody specificity is well taken. However, in our experimental setting, we have limited the interference of the mouse tissues by decreasing the fluorescence gain in our confocal analysis. Nevertheless, as correctly pointed out, we cannot exclude cross-reactivity. We could therefore validate the presence of human (tumor) cells in the lung tissues with one of the following approaches: (1) repeat the analysis on our paraffin embedded tissues, carrying out IHC/IF staining of lung sections by using a human-specific pan-Cytokeratin (ab234297, Abcam) and a human specific E-cadherin (ab40772, Abcam) antibody; (2) if the quality of the images would not be satisfactory and leave some doubts about the presence of human cell within the lung, we could alternatively, detect human Alu sequences by digital droplet PCR (ddPCR). A ddPCR based approach will indeed detect the presence of human DNA in mouse lungs, although this method will not help in quantifying human cells or visualizing metastases.

To avoid cross-reactivity between human and mouse species, we choose to stain mouse lung tissues with human antibody for PanCytokeratin (PanCK, ab234297 Abcam), being a well-known and highly used marker for identifying epithelial cancer cells. From IF analysis, we observed a very specific cell-membrane PanCK staining, allowing us to successfully detect metastatic nodules. Thus, we are confident that the staining/positivity of PanCK strongly support our in vivo data on the metastatic ability of prostate cancer cells (e.g. H&E staining) induced by the CAF/lactate conditioning and confirm the presence of human cancer cells in our metastasis experiments. These new data have been included as Appendix Figure S4C replacing the previous

2. The investigators show that CAF-CM contains higher lactate than PCa-CM and HPF-CM. If the presented data is a unique repeat and not a direct copy from the cited publications, I think it would be worth including in this manuscript to support the context.

These data have been obtained specifically for the revision, but very similar results have been already published in our previous works (PMID: 22850421, Can Res 2012). As these new data have been obtained in different CAFs lineages, obtained from different patients with respect to those published in 2012, we can include these in a new supplementary figure. In addition, concerning data published in 2012, we have some new data to include in the new supplementary figure, such as intratumoral lactate quantification, and the effect of MCT1 inhibitor on lactate uptake (suggested by reviewer#3).

We included these new data as Appendix Figure S2.

3. *I can't say that I am impressed with investigators argument for my point nr.3. Even considering the quantification, CAF-CM induction of P4HA1 is not as well neutralized by MCTi compared to LA induced P4HA1 in Figure 1e. The knockdown data in EV Figure 1g may be more convincing, however, the P4HA1 western blot is of poor quality, and doesn't look publication quality.*

We understand this point. One possible explanation is that the complexity of CAF-CM, containing a mixture of both metabolites and cytokines secreted by CAFs, could be accounted for this incomplete P4HA1 expression abrogation. To investigate this aspect, we propose to repeat the experiment by introducing a CAF-CM boiled sample \pm MCT1 inhibitor. A boiled CM will allow to eliminate the thermolabile molecules within the CM, including cytokines and chemokines. This approach will let emerge the role of metabolites, including lactate in the CAF-CM. Moreover, we will replace the EV Fig.1g with a better quality one.

We performed the experiment including boiled CAF-CM +/- MCT1i, to exclude the contribution of non-metabolic factors in the induction of P4HA1. We found that boiling CAF-CM does not impact on the P4HA1 induction, suggesting lactate as the main driver in inducing its overexpression. These data are included as new Figure 1E. We also replaced EV Fig.1g blot.

4. *The response to my point nr. 7 on STAT3 phosphorylation is not convincing. The examples of western blots shown in response to reviewers are not impressive. Personally, I would have problem trusting this data to show any real differences.*

We apologize for the low-quality image, likely due to the use of an old immunoblot acquisition instrument, due to a temporary breakdown of the device that we routinely use. The breakdown has been solved and we can repeat the experiment, again by introducing a CAF-CM boiled sample \pm DHB and/or P4HA1 silencing.

As above, we evaluated pSTAT3 levels also in boiled CAF-CM in both siP4HA1 and DHB-treated cells, and we found that pSTAT3 levels are slightly affected upon boiling CAF-CM, further corroborating the main role of metabolic cues (i.e. lactate) in sustaining STAT3 activation. These data are included as Appendix Figure S3C-D.

5. *The cartoon shown in Figure 4 to explain western blot data only shows half the experiment. It only shows the generation of cell free matrix, not what is plated on top and what is measured by western blot. I also think that the text needs to be revised to really clarify what point the authors want to make here.*

We will modify and revise the cartoon and the text accordingly, taking into consideration the Referee's concerns.

We modified the cartoon in the Figure 4B accordingly to the reviewer suggestions.

6. *In the response to my point nr. 6 on PCa spheres, the authors mention that sphere size is difficult to quantify using imaging software. I disagree, it should not be difficult to quantify sphere sizes. However, this is not a major point, so I leave it to the authors' discretion.*

We will quantify the size of spheres using a new software, since the pipeline for spheres quantification has now been optimized in the lab.

We included prostaspheres size quantification along their number in the main Figures.

Responses to Reviewer 2

The responses to Reviewer 2 seem to be fine overall. However, in the response to question nr. 2, the authors show experimental results (for example on lactate levels in different conditioned media as well as comparison with pyruvate and glutamine levels), but do not include these results in the paper. I would think the results would be a helpful addition to the paper, unless these exact plots are directly lifted from previous publications. The authors also provide further data in response to question nr. 3, such as intra-tumoral lactate levels to justify the concentration that they use for cell treatment in vitro. I would think these data would also be helpful to include in the manuscript (unless the exact plot is already published).

We thank the reviewer for having appreciated the revised manuscript. Also, we agree with the suggestion to include new experimental results, and we include intra-tumoral lactate levels, to better justify in vitro treatment with lactate we used (see Point 2 of Ref 1).

We added new data as Appendix Figure S2.

Referee #3:

This is a revised version of the previously submitted manuscript from Ippolito and colleagues. The authors have performed a detailed revision of the manuscript and included novel data. The authors have fully addressed all of my comments.

The Data Availability section, which is currently named as Data and code availability, is reserved for the primary datasets generated in the study. I note that the dataset GSE195639 was already published as a part of your 2022 study (PMID: 35135811). Please remove this section from the manuscript, and cite GSE195639 in the form of data citation in the text where appropriate. Please confirm that Figures 1a and b, which are derived from GSE195639, were not previously published in other publications. **Confirmed.**

Please see <https://www.embopress.org/page/journal/14693178/authorguide#referencesformat> for examples of data citation.

- Please rename the 'Authors' Disclosures' section as 'Disclosure Statement and Competing Interests' and place it after Acknowledgements. **Done.**
- Please remove the Authors' Contributions section from the manuscript. **Done.**
- Please make sure that the funding information is complete in the manuscript submission system. **Done.**
- Please resubmit Table EV1 as Dataset EV1. The file needs to be renamed as Dataset EV1 and the callout in the manuscript needs to be updated accordingly. Its legend the needs to be removed from the manuscript text and included in the same Excel file (as a separate tab/sheet). **Done.**
- The Appendix file needs a Table of Contents with page numbers. The figures should be renamed as Appendix Figure S1, etc. **Done.**
- Please rearrange the source data as one (zip) file per figure - i.e. please include the contents of the folder named 'uncropped WB Source Data' into the corresponding figure files. **Done.**
- During our routine figure checks, we noted a potential re-use of image between Figure 4D and Figure EV4 G (HPF-CM / siCOL1A1). Please clarify. **We apologize for this error in the image mounting. We included the correct figure in Fig.4D.**

The editorial requests below have been addressed.

- Our production/data editors have asked you to clarify several points in the figure legends:
 - o Please note that a separate 'Data Information' section is required in the legends of figures 1c-d, f; 2a-d; 3c-h; 4c-d; 5d-f; 6c, e; EV 1b-d; EV 2b, e; EV 4e-f, l-m; EV 5a-d.
 - o Please indicate the statistical test used for data analysis in the legends of figures 1b; EV 4e-f, l-m; EV 5b-d.
 - o Please note that in figures 1c-d, f; 2d; 3c-h; 5e-f; EV 3a, f; EV 4a, e-f; there is a mismatch between the annotated p values in the figure legend and the annotated p values in the figure file that should be corrected.
 - o Please note that the box plots need to be defined in terms of minima, maxima, centre, bounds of box and whiskers, and percentile in the legends of figures 1c; 2d; 5f; 6c; EV 1d; EV 3f.
 - o Please note that information related to n is missing in the legends of figures 1c, f; 6f-g; EV 2b; EV 4e-f, l-m; EV 5b-d.
 - o Please note that the scale bar is missing for figures 2b, d; 3g; 4d; 5e; EV 1c; EV 3e.
- Papers published in EMBO Reports include a 'synopsis' and 'bullet points' to further enhance discoverability. Both are displayed on the html version of the paper and are freely accessible to all readers. The synopsis includes a short standfirst summarizing the study in 1 or 2 sentences (max 35 words) that summarize the paper and are provided by the authors and streamlined by the handling editor. I would therefore ask you to include your synopsis blurb and 3-5 bullet points listing the key experimental findings.
- The synopsis image needs to be 550px wide and 300-600px high. When your synopsis image is resized accordingly, some of the labels are too small to read (please see attached). Please provide a synopsis image with larger labels.

Dear Paola,

Thank you for submitting your revised manuscript. It has now been seen by one of the original referees.

I apologize for the delay in getting back to you. Sometimes receiving the referee comments takes longer than anticipated.

As you can see, the referee finds that the study is significantly improved during revision and now recommends publication. However, I need you to address the points below before I can accept the manuscript.

- Please address the remaining minor concern of referee #1.
- We note the following regarding the funding information: missing: European Union, National Recovery and Resilience Plan, Mission 4 Component 2 - Investment 1.4 - National Center for Gene Therapy and Drugs based on RNA Technology - NextGenerationEU - Project Code CN00000041 - CUP B13C22001010001 (to EG) and Creation and strengthening of "innovation ecosystems", construction of "territorial R&D leaders" - TUSCANY HEALTH ECOSYSTEM (THE) NextGenerationEU - Project Code ECS_00000017 - CUP B83C22003920001 (to PC). Also, the following two need to be removed from the box and inserted as separate funders:
 - Swiss Cancer League (KLS-4899-08-2019) to Carlo V Catapano;
 - Fondazione Ticinese Ricerca sul Cancro to Carlo V Catapano.
- Please remove the legends of the Appendix figures from the manuscript text and insert them into the Appendix file. Every legend should follow its figure.
- The in text citations of Ippolito et al. 2022 and GSE195639 are correctly done. However, the reference list needs correction. Please cite Ippolito et al. 2022 and GSE195639 separately in the reference list as shown below:
Ippolito L, Comito G, Parri M, Iozzo M, Duatti A, Virgilio F, Lorito N, Bacci M, Pardella E, Sandrini G et al (2022) Gene Expression Omnibus GSE195639 (<https://www.ncbi.nlm.nih.gov/geo/query/acc.cgi?acc=GSE195639>) [DATASET]

Ippolito L, Comito G, Parri M, Iozzo M, Duatti A, Virgilio F, Lorito N, Bacci M, Pardella E, Sandrini G et al (2022) Lactate Rewires Lipid Metabolism and Sustains a Metabolic-Epigenetic Axis in Prostate Cancer. *Cancer Res* 82: 1267-1282

- Since the study involves human subjects, please include a statement in the Methods section that informed consent was obtained from all subjects and that the experiments conformed to the principles set out in the WMA Declaration of Helsinki and the Department of Health and Human Services Belmont Report.
- The synopsis image needs to be 550px wide and 300-600px high in jpeg, TIFF or png format. When your synopsis image is resized accordingly, the labels are too small to read (please see attached). Please provide a synopsis image with larger labels.

Thank you again for giving us to consider your manuscript for EMBO Reports, I look forward to your minor revision.

Kind regards,

Deniz

--

Deniz Senyilmaz Tiebe, PhD
Editor
EMBO Reports

Referee #1:

The authors have significantly revised the manuscript and addressed most of my concerns. They show an additional IF staining of lung metastases in their mouse model. The IF images are indeed more convincing than the H&E images. However, although the IF images show metastatic load that is in line with expected results, they are not quantified. The results that are quantified are based on the H&E staining. I could see two solutions here. Either to quantify the IF images and place in main figures instead of the H&E results, or to have a trained pathologist evaluate the H&E nodules and confirm that they are indeed metastases.

* Editor's comments *

- Please address the remaining minor concern of referee #1.

Please see the response below.

- We note the following regarding the funding information: missing: European Union, National Recovery and Resilience Plan, Mission 4 Component 2 - Investment 1.4 - National Center for Gene Therapy and Drugs based on RNA Technology - NextGenerationEU - Project Code CN00000041 - CUP B13C22001010001 (to EG) and Creation and strengthening of "innovation ecosystems", construction of "territorial R&D leaders" - TUSCANY HEALTH ECOSYSTEM (THE) NextGenerationEU - Project Code ECS_00000017 - CUP B83C22003920001 (to PC). Also, the following two need to be removed from the box and inserted as separate funders:

- Swiss Cancer League (KLS-4899-08-2019) to Carlo V Catapano;
- Fondazione Ticinese Ricerca sul Cancro to Carlo V Catapano.

Done.

- Please remove the legends of the Appendix figures from the manuscript text and insert them into the Appendix file. Every legend should follow its figure.

Done.

- The in text citations of Ippolito et al. 2022 and GSE195639 are correctly done. However, the reference list needs correction. Please cite Ippolito et al. 2022 and GSE195639 separately in the reference list as shown below:

Ippolito L, Comito G, Parri M, Iozzo M, Duatti A, Virgilio F, Lorito N, Bacci M, Pardella E, Sandrini G et al (2022) Gene Expression Omnibus GSE195639
(<https://www.ncbi.nlm.nih.gov/geo/query/acc.cgi?acc=GSE195639>) [DATASET]

Ippolito L, Comito G, Parri M, Iozzo M, Duatti A, Virgilio F, Lorito N, Bacci M, Pardella E, Sandrini G et al (2022) Lactate Rewires Lipid Metabolism and Sustains a Metabolic-Epigenetic Axis in Prostate Cancer. Cancer Res 82: 1267-1282

Done.

- Since the study involves human subjects, please include a statement in the Methods section that informed consent was obtained from all subjects and that the experiments conformed to the principles set out in the WMA Declaration of Helsinki and the Department of Health and Human Services Belmont Report.

Done (lane 384-387).

- The synopsis image needs to be 550px wide and 300-600px high in jpeg, TIFF or png format. When your synopsis image is resized accordingly, the labels are too small to read (please see attached). Please provide a synopsis image with larger labels.

We provided a synopsis image with more readable labels.

* Reviewer's comments *

Referee #1:

The authors have significantly revised the manuscript and addressed most of my concerns. They show an additional IF staining of lung metastases in their mouse model. The IF images are indeed more convincing than the H&E images. However, although the IF images show metastatic load that is in line with expected results, they are not quantified. The results that are quantified are based on the H&E staining. I could see two solutions here. Either to quantify the IF images and place in main figures instead of the H&E results, or to have a trained pathologist evaluate the H&E nodules and confirm that they are indeed metastases.

We quantified the IF images accordingly, and we replaced the H&E images (Appendix Figure S4) with IF plus quantification graph as main Figure 6D-E.

Prof. Paola Chiarugi
University of Florence
Biochemical Sciences
Viale Morgagni 50
Florence, Italy 50134
Italy

Dear Paola,

Thank you for submitting your revised manuscript. I have now looked at everything and all is fine. Therefore, I am very pleased to accept your manuscript for publication in EMBO Reports.

Congratulations on a nice work!

Kind regards,

Deniz

--

Deniz Senyilmaz Tiebe, PhD
Editor
EMBO Reports
